# Empowering Education with Generative Artificial Intelligence Tools: Approach with an Instructional Design Matrix

Lena Ivannova Ruiz-Rojas [1], Patricia Acosta-Vargas [2,3,*], Javier De-Moreta-Llovet [4]
and Mario Gonzalez-Rodriguez [2,5,*]

1   Departamento de Ciencias Humanas y Sociales, Universidad de Las Fuerzas Armadas,
    Sangolquí 170550, Ecuador; liruiz@espe.edu.ec
2   Intelligent and Interactive Systems Laboratory, Universidad de Las Américas, Quito 170125, Ecuador
3   Carrera de Ingeniería Industrial, Facultad de Ingeniería y Ciencias Aplicadas, Universidad de Las Américas,
    Quito 170125, Ecuador
4   Facultad de Derecho, Universidad Complutense de Madrid, 28040 Madrid, Spain; jamoreta@ucm.es
5   Carrera de Ingeniería de Software, Facultad de Ingeniería y Ciencias Aplicadas, Universidad de Las Américas,
    Quito 170125, Ecuador
*   Correspondence: patricia.acosta@udla.edu.ec (P.A.-V.); mario.gonzalez.rodriguez@udla.edu.ec (M.G.-R.)

**Abstract:** This study focuses on the potential of generative artificial intelligence tools in education, particularly through the practical application of the 4PADAFE instructional design matrix. The objective was to evaluate how these tools, in combination with the matrix, can enhance education and improve the teaching–learning process. Through surveys conducted with teachers from the University of ESPE Armed Forces who participated in the MOOC course "Generative Artificial Intelligence Tools for Education: GPT Chat Techniques", the study explores the impact of these tools on education. The findings reveal that generative artificial intelligence tools are crucial in developing massive MOOC virtual classrooms when integrated with an instructional design matrix. The results demonstrate the potential of generative artificial intelligence tools in university education. By utilizing these tools in conjunction with an instructional design matrix, educators can design and deliver personalized and enriching educational experiences. The devices offer opportunities to enhance the teaching–learning process and tailor educational materials to individual needs, ultimately preparing students for the demands of the 21st century. The study concludes that generative artificial intelligence tools have significant potential in education. They provide innovative ways to engage students, adapt content, and promote personalized learning. Implementing the 4PADAFE instructional design matrix further enhances the effectiveness and coherence of educational activities. By embracing these technological advancements, education can stay relevant and effectively meet the digital world's challenges.

**Keywords:** methodology; MOOC; digital resources; teaching strategies

## 1. Introduction

In today's digital era, education is undergoing a paradigm shift driven by technological advancements. The integration of generative artificial intelligence (AI) tools and instructional design matrices represents an innovative and promising approach to addressing the evolving needs of modern education. By harnessing the power of AI, educators can leverage personalized learning experiences, adaptive content generation, and real-time support for students. Using an instructional design matrix adds structure and coherence to the educational process, ensuring alignment with learning objectives and enhancing the effectiveness of teaching strategies. This combined approach improves student engagement and motivation and offers educators new opportunities to create dynamic and inclusive virtual classrooms. By exploring these technologies' potential benefits and implications, this paper aims to inspire educators and institutions to embrace generative AI tools and instructional design matrices as transformative tools in empowering education.

Education has undergone a significant transformation in today's rapidly evolving digital landscape. The integration of generative artificial intelligence (AI) tools and instructional design matrices has revolutionized the way learning activities are conceived and executed [1,2]. The application of AI in education holds immense potential, offering new possibilities for personalized learning experiences and adaptive teaching approaches. Simultaneously, the increasing digitization of society has propelled the prominence of artificial intelligence with its ability to automate tasks, analyze vast amounts of data, and provide predictive insights that have far-reaching implications across various domains [3].

The stated problem of enhancing education using generative artificial intelligence tools in a practical approach directly relates to the research question: "How can education be enhanced using generative artificial intelligence tools in a practical approach, applying the 4PADAFE instructional design matrix?" 4PADAFE [4] stands for Academic Project, Strategic Plan, Instructional Planning, Instructional Material Production (4P), Teaching Action (AD), Formative Adjustments (AF), and Evaluation (E). The research question seeks to specifically inquire into the possibilities and benefits of generative artificial intelligence tools, combined with the 4PADAFE matrix, to enhance the teaching–learning process. The aim is to explore how these tools can personalize learning, provide immediate feedback, adapt educational materials, and promote the development of key skills in students. By analyzing and evaluating the potential of these tools in a practical approach, we seek to identify effective strategies for their implementation in education to optimize the student experience and strengthen the quality of education in a digitalized and constantly evolving environment.

Another problem is the lack of knowledge on the part of teachers of generative artificial intelligence tools and systematic processes for designing micro-curricular activities that guide the development and construction of massive virtual learning classrooms; this would enable teachers to manage activities and design digital resources with artificial intelligence with innovative educational strategies that facilitate the creation of massive virtual classrooms [1].

The general objective of this research has been to analyze and evaluate the potential of generative artificial intelligence tools in the educational context, focusing on the practical application of the 4PADAFE instructional design matrix.

Based on the stated objectives, the proposed scientific or working hypothesis to be contrasted or demonstrated in this study is that applying generative artificial intelligence tools in the educational environment, using the 4PADAFE instructional design matrix, positively impacts the teaching–learning process.

Based on the statement above, this combination of tools and practical approach is expected to improve academic results and student engagement in learning, promoting more efficient, effective, and personalized education.

The present study shows the results of the implementation of the 4PADAFE methodology; during the development of the course "Generative Artificial Intelligence Tools for Education. ChatGPT Techniques", teachers were given the task of designing a teaching unit using both the generative artificial intelligence tools (IAG) learned in the course and the 4PADAFE instructional design matrix.

In practice, the teachers demonstrated a solid understanding and application of the 4PADAFE methodology and the generative artificial intelligence tools. First, they conducted detailed planning, identifying the specific learning objectives they wanted to achieve in their teaching unit. From there, they designed micro-curricular activities that aligned with the principles of the 4PADAFE matrix.

The activities proposed by the teachers involved using IAG tools at different stages of the educational process. For example, they designed initial activities that took advantage of the content generation capabilities of IAG tools to present the contents of a subject attractively. They also designed interactive activities where students interacted with different IAG tools such as chatPDF.com, You.com, chatbots, and virtual assistants to solve problems, receive feedback, or explore new ideas.

In addition, teachers used IAG tools to evaluate student performance more efficiently and effectively. These tools allowed them to analyze student-generated responses, assess comprehension, and provide real-time personalized feedback.

In summary, the teachers demonstrated a solid command of the 4PADAFE methodology and generative artificial intelligence tools. They planned and designed their micro-curricular activities effectively using IAG tools, leveraging their capabilities to enhance content presentation, promote student–teacher interaction, and streamline learning assessment. These combined approaches resulted in dynamic and enriching teaching units that fostered active student participation and facilitated the achievement of the learning objectives.

In conclusion, current technological educational trends emphasize the use of generative artificial intelligence tools, which allow the creation of personalized educational content tailored to the needs of students. In addition, implementing an instructional design matrix provides a structured guide for developing micro-curricular activities, ensuring coherence and quality in the educational process.

On the other hand, implementing an instructional design matrix in the educational process has become a fundamental practice to guarantee quality and coherence in the design of co-curricular activities. An instructional design matrix provides a structure and a clear guide for the development of the activities, considering the educational objectives, the contents, the teaching strategies, the evaluations, and the necessary resources. By following an instructional design matrix, educators can ensure that activities align with learning objectives and promote an effective and meaningful educational process [4].

The evolution of generative artificial intelligence raises the need to reconsider the teaching–learning process since its impact extends to the trend of adaptive education. This trend can potentially have a significant impact on conventional learning approaches. As new and better AI-based applications are developed, the new curricula will likely become more responsive and versatile enough to quickly adapt to the latest and most efficient ways of approaching education in the present century [5].

This article will explore generative artificial intelligence tools, emphasizing some of them, such as ChatGPT [6], Fliki Ai [7], You.com [8], Studio.Ai [9], Chat Pdf.Com [10], Leonardo AI [11], and Humata.ai [12]. These tools use advanced algorithms to create and generate educational content in an automated way, providing new opportunities for personalized learning and adaptation to students' individual needs [13]. The impact of generative artificial intelligence on the personalization of education will be examined, as well as how implementing an instructional design matrix can improve the effectiveness and coherence of the educational process. In addition, examples of good practice will be presented, and the future implications of these technological educational trends will be discussed.

This study contributes by exploring the integration of generative AI tools and instructional design matrices as transformative tools in education. It highlights AI's potential to personalize learning, generate adaptive content, and provide real-time support. The instructional design matrix adds structure and coherence, enhancing teaching strategies and student engagement. Implementing AI tools and the 4PADAFE matrix in a course demonstrated their practical application, enabling teachers to design interactive activities, improve content presentation, and streamline assessment. This integration resulted in dynamic teaching units that fostered active student participation and the achievement of learning objectives.

This paper is organized into several sections. In Section 2, readers are introduced to generative artificial intelligence tools and their application in implementing MOOC courses using the 4PADAFE methodology. Section 3 describes the research methodology used to harness artificial intelligence in developing virtual course content. The analysis findings, which used artificial intelligence and the 4PADAFE methodology in implementing MOOC courses, are presented in Section 4. A comprehensive discussion of the results is provided in Section 5, while Section 6 focuses on the implications of the practices. The conclusions drawn from the research are presented in Section 7.

## 2. Generative Artificial Intelligence Tools and the 4PADAFE Instructional Matrix

Generative artificial intelligence tools automatically generate personalized educational content using advanced algorithms. These tools analyze student data and create tailored materials, enhancing active participation and motivation for learning. They allow the creation of personalized educational content based on individual needs and preferences [14]. A Scopus search on "generative artificial intelligence tools in education" yielded 39 scientific articles.

The related keywords formed two main clusters: one focused on artificial intelligence, ChatGPT, education, and learning systems; the other on deep learning, human, and machine learning. This graphical representation helps identify relevant research areas for future studies in generative artificial intelligence tools in education. Figure 1 shows a neural network with two main groups generated with VOSviewer [15].

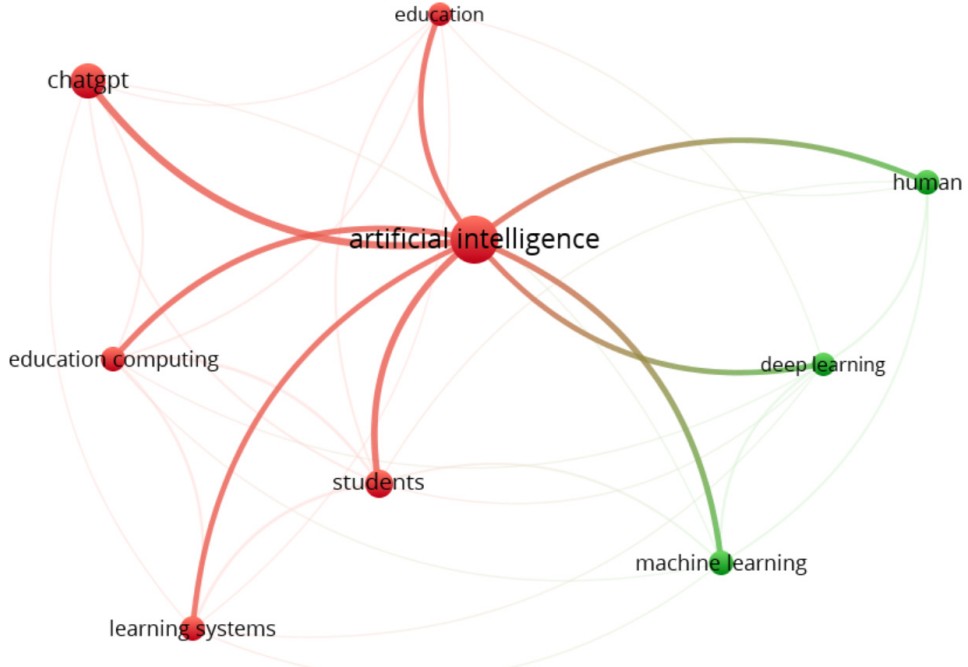

**Figure 1.** Graphical representation of related keywords generated with VOSviewer.

The article [16] points out that generative artificial intelligence (AI) offers transformative potential in education. This exploratory paper applies a self-study methodology to investigate this technology. The results of ChatGPT are usually aligned with the central themes of the study, and for this reason, it is essential that educators model responsible use of ChatGPT, prioritize critical thinking, and be clear about expectations. ChatGPT is likely helpful for educators in designing science units, rubrics, and quizzes.

Another study [17] suggests that future teachers should have the opportunity to access courses focused on applying artificial intelligence and integrating technology in the classroom as an integral part of their initial training. However, students need to recognize that they have primary responsibility as learners and should not rely solely on AI applications, such as ChatGPT, to fulfill their educational role.

One of the problems identified is the lack of knowledge about generative artificial intelligence tools and systematic processes for designing co-curricular activities that guide the development and construction of massive virtual learning classrooms; these allow teachers to manage activities and digital resources with artificial intelligence and educational strategies to create their virtual classrooms. In this study, the 4PADAFE methodology makes it possible to align the learning objectives, contents, activities, and evaluations, guaranteeing the coherence and quality of the instructional design.

The supply of "smart" applications with possible educational and/or academic uses is experiencing constant growth. New options are added daily to the wide range of generative AI tools available. Websites and specialized portals such as Futurepedia or All Things AI provide access to various tools. These resources allow educators and academics to explore and harness the potential of generative artificial intelligence to enrich their pedagogical practices and promote more interactive and personalized learning [18].

The influence of generative artificial intelligence in university education has been significant in recent years. The ability of artificial intelligence tools to automatically generate educational content has transformed how education is delivered and accessed at the university level [19]. Some relevant artificial intelligence tools and their importance in university teaching are described below [20].

ChatGPT is an artificial intelligence tool that uses generative language models to interact and answer questions conversationally [6]. This tool allows teachers to use it as a virtual assistant to provide answers to student queries, offer additional information, and provide personalized support in real time.

Fliki AI is an artificial intelligence tool [7] designed to create educational content. It allows for generating didactic materials, such as interactive presentations, quizzes, and adaptive learning activities. University faculty can use Fliki AI to develop high-quality, personalized educational resources tailored to the needs of their students.

You.com is a search engine that combines search results, applications, and shortcuts to present information in an organized and easy-to-use way.

Aistudio.com is an artificial intelligence video generation platform that uses an AI avatar. It allows to generate realistic AI videos quickly and efficiently [21].

Chat Pdf.com is a system based on artificial intelligence that allows us to "read" and synthesize the most important ideas and returns a complete summary of any document in PDF format. One of its features is that it understands any language and can reply in the chosen language. It is used to propose co-curricular activities; we can use the tool to summarize documents and then use the summaries to pose discussion questions or assign writing activities.

Leonardo AI is an artificial intelligence tool that uses computer vision and machine learning to analyze images and videos. In the university context, teachers can use Leonardo AI to enhance the teaching experience, such as identifying objects in scientific experiments or interpreting medical images in the medical field.

Humata.ai is an artificial intelligence platform that uses machine learning algorithms to analyze course content and provide personalized recommendations to college students. Teachers can use Humata.ai to adapt the content and teaching methodology to the needs of the students, improving the learning experience.

These artificial intelligence tools are transforming college education by providing new opportunities to personalize learning, improve the quality of content, and facilitate interaction between students and faculty. They allow educators to adapt to the needs of students, offering a more personalized and practical approach to teaching and learning [22].

Properly implementing these tools requires careful planning and effective integration into the instructional design. University professors must receive adequate training on using these tools effectively and maximizing their potential to benefit students.

The present methodology proposes systematic cyclical processes to develop a massive virtual course efficiently. It should allow the construction of didactic material and the development of activities in an orderly and structured manner while responding to the application of innovative pedagogical bases and learning resources for a MOOC. Its interactive component facilitates training adjustments, which aligns with the idea that, even when everything has been planned, adjustments are possible to accommodate the specific group one happens to be working with [23]. Figure 2 presents an infographic of the 4PADAFE methodology.

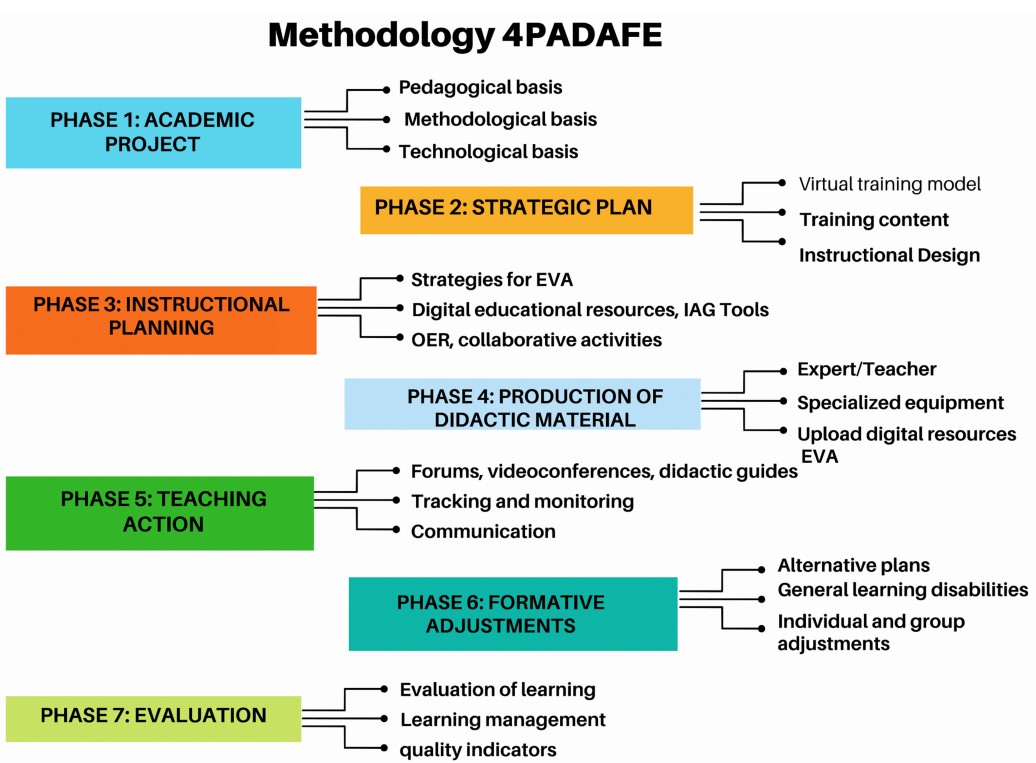

**Figure 2.** 4PADAFE Methodology.

The methodology proposed in this study is based on integrating the principles and combining socio-constructivist and connectivist models. It focuses on effective communication and engages students in open learning and reflective inquiry [23].

The 4PADAFE methodology, which consists of seven phases, is a reference for this study. Phase 1 involves the eLearning plan and the academic teaching project, followed by the strategic plan (Phase 2), instructional planning (Phase 3), production of didactic material (Phase 4), teaching action (Phase 5), training adjustments (Phase 6), and evaluation (Phase 7) [2].

The concept of "techno-instructional or techno-pedagogical design" recognizes the inseparable connection between the technological and pedagogical dimensions in virtual training. The technological dimension involves selecting appropriate tools, such as virtual platforms, software applications, and multimedia resources, while the pedagogical dimension requires understanding the target audience, analyzing objectives or competencies, developing and implementing content, planning activities, and evaluating processes and results [24].

These concepts and methodologies serve as a framework for instructional design in virtual training to effectively integrate technology and pedagogy to enhance the learning experience.

The development starts with Phase 2 of the instructional design matrix to address the pedagogical dimension. Figure 3 outlines the following key points to be considered:

(a) Select a subject or area that aligns with the desired profile. (b) Choose the specific units to be developed. (c) Identify one or two topics per unit for developing learning, didactic, and digital strategies. (d) Clearly state the objective for each unit. (e) Consider the topics that will be covered in each unit. (f) Define the expected learning outcome for each unit.

These guidelines provide a systematic approach to designing the pedagogical aspect of the instructional design matrix, ensuring a structured and effective learning process.

| INSTRUCTIONAL DESIGN MATRIX METHODOLOGY 4PADAFE | | | |
|---|---|---|---|
| **Name of the module/subject/course: Course Educational tools with AI**<br>**Project development time: 4 weeks**<br>**Link (read): https://repositorio.grial.eu/handle/grial/2138** | | | |

| TARGET PUBLIC DIAGNOSIS | | | |
|---|---|---|---|
| **Age:** | **Education level:** | **Learning strategies:**<br>**Motivation: New skills** | **Technology experience:**<br>**Experience virtual courses:** |

| FASE 2: PLAN ESTRATÉGICO | | | |
|---|---|---|---|
| **UNITS** | **Competencies/Learning Outcomes** | **Goals** | **Contents / TOPICS** |
| **I: An educational trend with AI** | | | TOPIC 1:<br>TOPIC 2:<br>TOPIC 3: |

| INSTRUCTIONAL DESIGN MATRIX METHODOLOGY 4PADAFE | | | | | | | | | | |
|---|---|---|---|---|---|---|---|---|---|---|
| **PHASE 2: STRATEGIC PLAN** | | | | **PHASE 3: INSTRUCTIONAL PLANNING STUDENT ACTIVITIES** | | | | | **PHASE 5: TEACHING ACTION** | **PHASE 7: EVALUATION** |
| **UNIT**<br>(Develop unit) | **WEEKS AND DATES** | **RESULTS AND LEARNING MODELS** | **CONTENTS** | **SYNCHRONOUS** | **ON LINE ASYNCHRONOUS** | **OFFLINE HOURS OF DEDICATION** | | | **STUDY MATERIAL USING DIGITAL RESOURCES** | **DESCRIBE ACTIVITIES IN THE DIDACTIC GUIDES, WITH EVALUATION RUBRIC** | **DESCRIBE THE TOOL TO EVALUATE THE ACTIVITY** |
| **I Educational trends with AI**<br>April 1 to 16, 2023 | **MONDAY:**<br>April 3, 2023<br><br>11:00 – 12:00 | **Topic 1**<br>**Speech and Debate in the Classroom** | **- Know the different methodologies**<br>**- Learning the different formats of …..**<br>**- Creation of…** | CD: watch an introductory video<br><br>Academic forum | **C.L.A. 1**<br>**Reading the PDF material ..**<br><br>**C.L.A. 2**<br>**Watch video** | C.L.A. 1<br>30 min | C.L.A. 2<br>30 min | P.C.<br>30 min | Create an introductory video of the whole week, with the calendar on screen, explaining activities from Monday to Sunday using OBS or others.<br>Design a video for each topic<br>Create tutorials for each tool<br>Design an | 1. View the presentation in PWP<br>2. Carry out a debate in the classroom in teams.<br>3. Participate actively in defending a position with reasoned arguments. | Diagnostic evaluation Develop ten diagnostic questions, multiple choice of each UNIT.<br><br>Every week an evaluation |
| **ACTIVITIES BY COMPONENT** | Number of hours of the teaching component (SYNCHRONOUS ACTIV.), CT: 2 H<br>Number of hours of the PAC application and experimentation practice components: 2 H<br>Number of hours of autonomous learning (ASYNCHRONOUS ACTIV.) CLA: 6 H<br>Total number of hours: 10 | | | | | | | | | |

**Figure 3.** 4PADAFE instructional design matrix methodology for designing a virtual course.

In the context of virtual training, various generative artificial intelligence tools, including Chat GPT, Fliki Ai, You.com, Aistudio, Chat Pdf.com, Leonardo Ai, and Humata.ai, among others, coupled with the utilization of an instructional design matrix, enable the enhancement and personalization of learning experiences. These tools and methodologies offer more effective and enriching educational experiences for students.

The 4PADAFE methodology, which supports the design of virtual classrooms, incorporates technological resources and generative artificial intelligence tools to plan training activities. Technology serves as a means to fulfill academic planning, and teachers must select appropriate technological strategies to ensure the success of the learning process. Educators must be knowledgeable about and proficient in applying AI tools and digital resources to create digital content. This development occurs in parallel with Phase 3, Phase 5, and Phase 7 of the instructional design matrix, as illustrated in Figure 3.

When designing synchronous and asynchronous activities, it is advisable to utilize the instructional design matrix of the 4PADAFE methodology. Consider the following guidelines: (1) Employ captivating video classes that create a "WOW" effect, capturing and maintaining students' attention. (2) Utilize diverse learning strategies tailored to the specific target audience. (3) Choose strategies that align with the subject matter, adjusting the complexity level as students progress in their knowledge. (4) Apply motivational strategies, as motivation and learning are interrelated with academic performance. (5) Emphasize that the teacher's effectiveness lies in their knowledge of the subject matter and their mastery of various didactic strategies and generative artificial intelligence tools. (6) Utilize learning strategies and AI-powered tools to promote students' gradual autonomy.

Teachers must define innovative working methods by leveraging generative artificial intelligence tools and digital resources to enhance the learning experience.

Figure 4 describes the building of micro-curricular activities with the 4PADAFE instructional design matrix. It shows how to design a Phase 2 strategic plan. In this section, the contents are structured into didactic units; the unit is organized by weeks, and each subject's specific and generic competencies, objectives, and learning outcomes are identified.

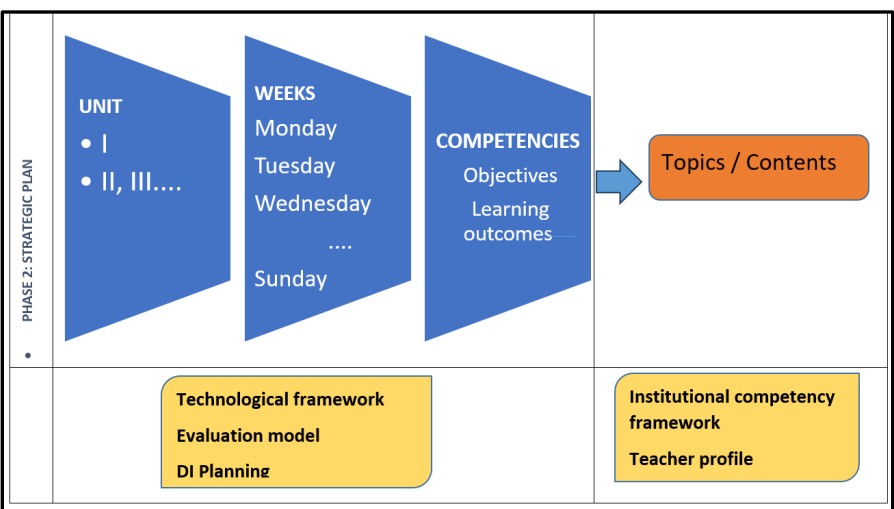

**Figure 4.** 4PADAFE methodology for creating a virtual course, phase 2.

Figure 5 shows the proposed synchronous (online) and asynchronous (offline) activities and how to transform a boring class into a fun, motivating class with feedback. It should be noted that to automate the co-curricular activities, an LMS, a virtual learning environment, is needed.

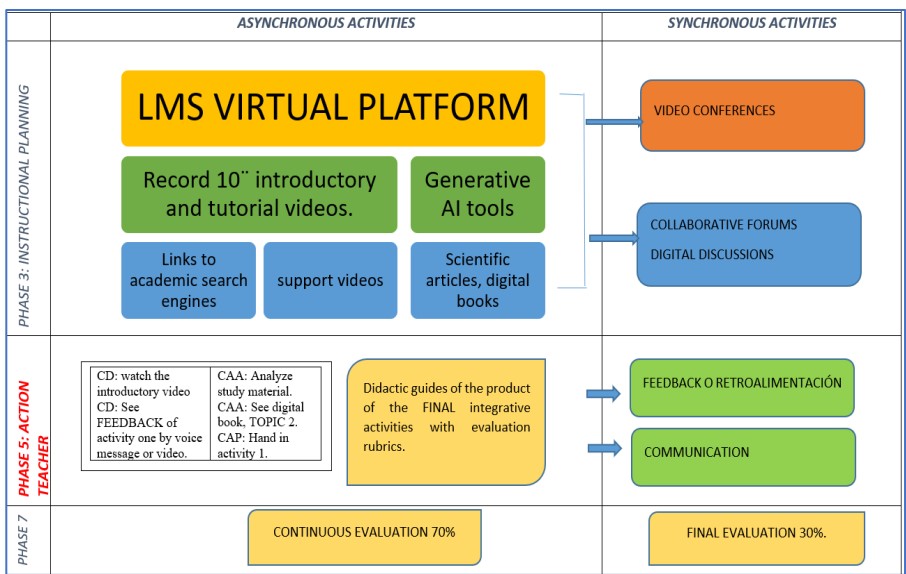

**Figure 5.** 4PADAFE methodology for creating a virtual course, phase 3.

Creating a virtual classroom within a virtual learning environment involves following the instructional design matrix described in Figure 5. The matrix emphasizes the importance of strategic planning (Phase 2), instructional planning (Phase 3), teaching actions (Phase 5), and evaluation (Phase 6).

Phase 2 focuses on developing micro-curricular planning for each unit, including defining competencies, objectives, contents, and learning outcomes every week. Phase 3 builds upon Phase 2 and involves creating study materials using digital resources, interactive multimedia technologies, and virtual learning environments. Synchronous and asynchronous

learning activities are planned, with the learning organization determined by the number of hours allocated to the teaching component (CD), application and experimentation practices (CPA), and autonomous learning (CAA).

CD's teaching component encompasses direct teacher interaction, collaborative learning activities, video lectures, forums, discussions, joint exercises, and feedback. Digital resources such as videos, videoconferences, podcasts, presentations, concept maps, and digital books are utilized. The application and experimentation practices component involves practical or laboratory activities complementing the study material. Autonomous learning consists of activities independently developed by students under teacher guidance. These components lead to Phase 5, where integrative activities or final products are proposed to achieve the training objectives. Phase 4 involves designing didactic materials based on micro-curricular planning, while Phase 6 defines the number of assessments per week.

Various educational resources can be employed to enhance student learning, including audiovisual materials, which should be concise (no longer than 5 min), with high image and sound quality, and accompanied by a well-developed script. Combining different types of educational resources can increase student interest and motivation. It is essential to ensure compliance with accessibility requirements and creative commons licenses for these resources.

In online training, teachers fulfill new roles supported by generative artificial intelligence and digital strategies. This methodology includes using digital tools and resources to design engaging educational materials, conducting synchronous and asynchronous tutoring sessions, facilitating academic forums, providing guidance on technology usage, monitoring the learning process, evaluating student progress, and providing feedback. Continuous evaluation processes are carried out using formal and informal tools and evaluation rubrics.

These methodologies and strategies aim to optimize the online learning experience, recognizing virtual education's unique characteristics and advantages.

Micro-curricular planning in virtual environments requires teachers to possess pedagogical knowledge, didactic skills, and digital expertise [25]. When implementing artificial intelligence activities, the following steps are suggested [19,26]:

Identify areas where artificial intelligence can be beneficial, such as providing personalized feedback or analyzing student work for progress assessment.

Explore available AI tools and resources suitable for classroom implementation.

Acquire proficiency in using AI tools through learning and training.

Design activities that incorporate artificial intelligence, promoting collaborative work and critical thinking.

Test the activities with students to ensure their effectiveness.

In conclusion, achieving a coherent micro-curricular planning and didactic sequence requires teachers to be well-prepared in the subject matter and possess pedagogical knowledge, didactic skills, and proficiency in virtual platforms, digital environments, and generative artificial intelligence tools [25].

## 3. Materials and Methods

This study employed a quantitative research design to investigate the impact of generative artificial intelligence tools and the implementation of an instructional design matrix on the construction of massive MOOC virtual classrooms.

The participants in this study were 42 teachers; the selection process involved purposive sampling to ensure representation from different disciplines and levels of teaching experience. All participants had experience using generative artificial intelligence tools in their educational practices.

In our study, we used a purposive sampling approach to select participants, ensuring representation from different disciplines and levels of teaching experience. While we recognize that a larger sample size could increase the generalizability of the results, it is

also essential to keep in mind the nature of our study, which focused on the specific context of massive online classrooms (MOOCs).

MOOCs are online learning environments that allow the participation of large numbers of students from diverse geographic locations and educational backgrounds. Since our study focused on the impact of generative artificial intelligence tools and instructional design matrices in this specific context, we felt that a sample of 42 teachers was adequate to capture various relevant experiences and perspectives.

The course to which the methodology was applied had a total of 47 teachers, of which 42 responded; when using the formula with a known universe size, with a margin of error of 5% and a confidence level of 95%, the sample of 42 teachers is within the allowed range.

A quantitative survey administered to participants was used for data collection. The survey comprised a questionnaire on generative artificial intelligence tools, including ChatGPT, Humata.ai, and other identified devices. Participants were also asked about using the 4PADAFE instructional design matrix in their teaching practices. The survey was administered online, and participants provided their responses anonymously. The survey dataset is available in the Mendeley repository [27].

Several steps were taken to ensure the survey's accuracy and validity. First, a peer validation process was used, where experts reviewed the questionnaire and provided comments and suggestions for improvement. Adjustments and revisions were made based on these comments to ensure the questions were clear and relevant.

In addition, reliability measures were used to assess the internal consistency of the items in the questionnaire. Cronbach's alpha coefficient was applied in the SPSS application, and Table 1 shows the reliability of the measurement scales used in the questionnaire. A Cronbach's alpha coefficient of 0.978 indicates a higher internal consistency of the questions in each scale. The value obtained is very positive and suggests a high internal consistency of the items used in the questionnaire. This value is indicative of the high internal consistency of the items used in the questionnaire.

**Table 1.** Cronbach's alpha coefficient was calculated with SPSS statistical analysis software.

| Cronbach's Alpha | Cronbach Alpha Based on Standardized Items | N of Elements |
|---|---|---|
| 0.978 | 0.979 | 42 |

Regarding the participant selection process, purposive sampling was used to ensure representation from different disciplines and levels of teaching experience. Forty-two university professors with experience using generative artificial intelligence tools in their educational practices were selected.

Quantitative data obtained from the surveys were analyzed using descriptive statistics. The frequency and percentage of tool use were calculated, as well as the use of instructional design matrices, to identify patterns and trends among participants.

It was found that a small percentage of teachers show total resistance to the use of generative artificial intelligence tools and do not recognize their relevance in the learning process. This may be due to a lack of opportunities to experiment with these tools and a lack of alignment with current educational trends. It is critical to address these concerns and resistance by providing training and support for teachers to understand the potential benefits of these tools in education.

It is important to recognize certain limitations of this study. First, the sample size was relatively small, consisting of teachers from a specific region and academic disciplines. Generalization to larger populations may be limited. Second, the study focused on using specific generative artificial intelligence tools and instructional design matrices, potentially excluding other tools and methodologies. Finally, the study relied on self-reported data, which may be subject to response bias.

Despite these limitations, this study provides valuable insights into the impact of generative artificial intelligence tools and instructional design matrices in the context of MOOC virtual classrooms. The findings contribute to the existing literature and should inform future educational research and practice.

## 4. Results

The results revealed that generative artificial intelligence tools, such as ChatGPT, Humata.ai, and other identified tools, were widely utilized by the participating teachers in their educational practices. Figure 6 presents the AI tools that teachers most frequently utilized among the participants. ChatGPT emerged as the most popular tool, with a usage rate of 95.2%. Humata.ai followed at 31%, Chat PDF.com at 28.6%, Studio.AI at 26.2%, Leonardo AI at 16.7%, Tome AI and You.com at 14.3% each, and finally, Fliki AI at 11.9%. These statistics demonstrate educators' widespread adoption and utilization of these generative AI tools.

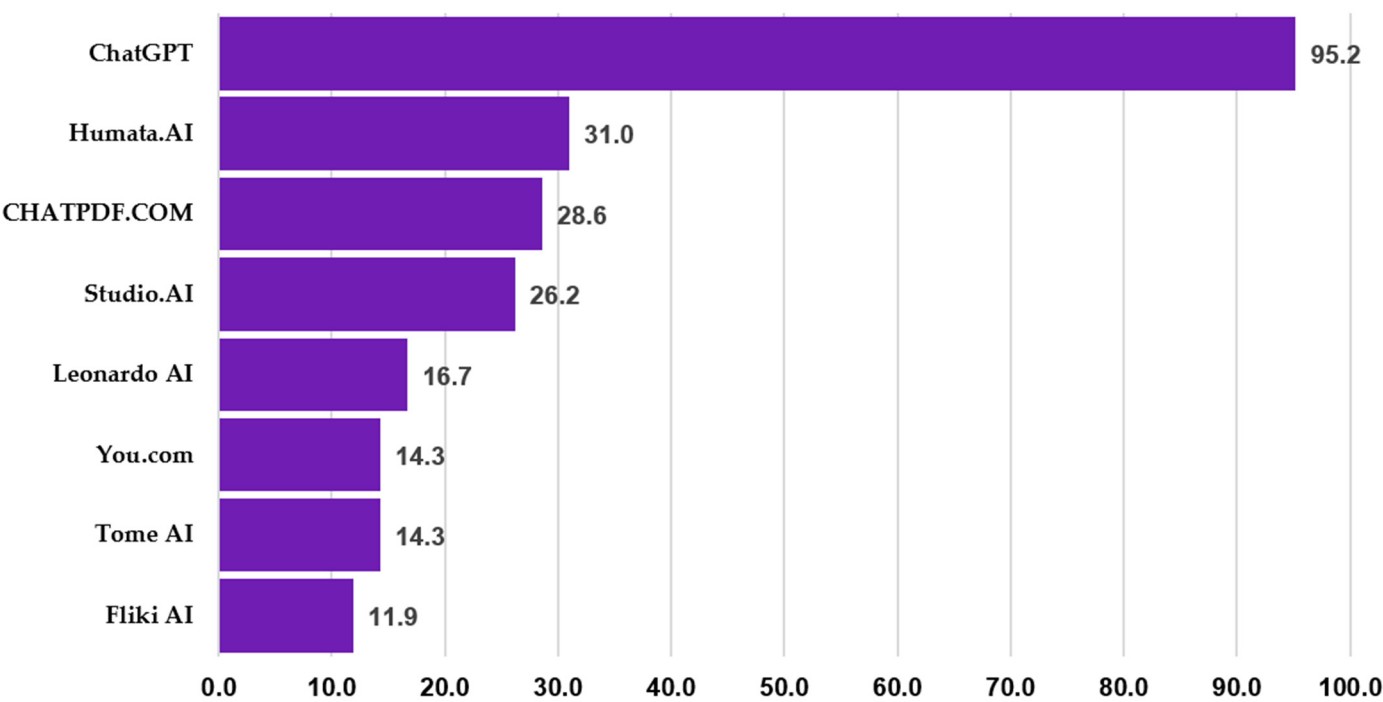

**Figure 6.** Summary of the generative AI tools most frequently utilized by teachers.

These tools were primarily employed as virtual assistants, providing personalized responses, answering student queries, and offering real-time support. Using generative artificial intelligence tools significantly contributed to student engagement and participation in virtual classrooms.

Figure 7 shows that after conducting the analysis, the questions related to ChatGPT, 4PADAFE, and MOOC that received more positive responses on the Likert scale, with values between 4 and 5 indicating "Agree" and "Strongly agree", respectively, were grouped. These responses were classified according to the age of the participants, and it was observed that individuals aged 47 to 57 responded most positively, followed by those aged 36 to 46 and then those aged 25 to 35. The responses reflect a positive perception of how artificial intelligence tools enhance the educational experience. Additionally, it is considered that ChatGPT and the 4PADAFE matrix can help teachers save time in preparing materials and educational resources for MOOC courses. Lastly, it is believed that ChatGPT can facilitate the exploration of complex topics in the classroom more easily for students.

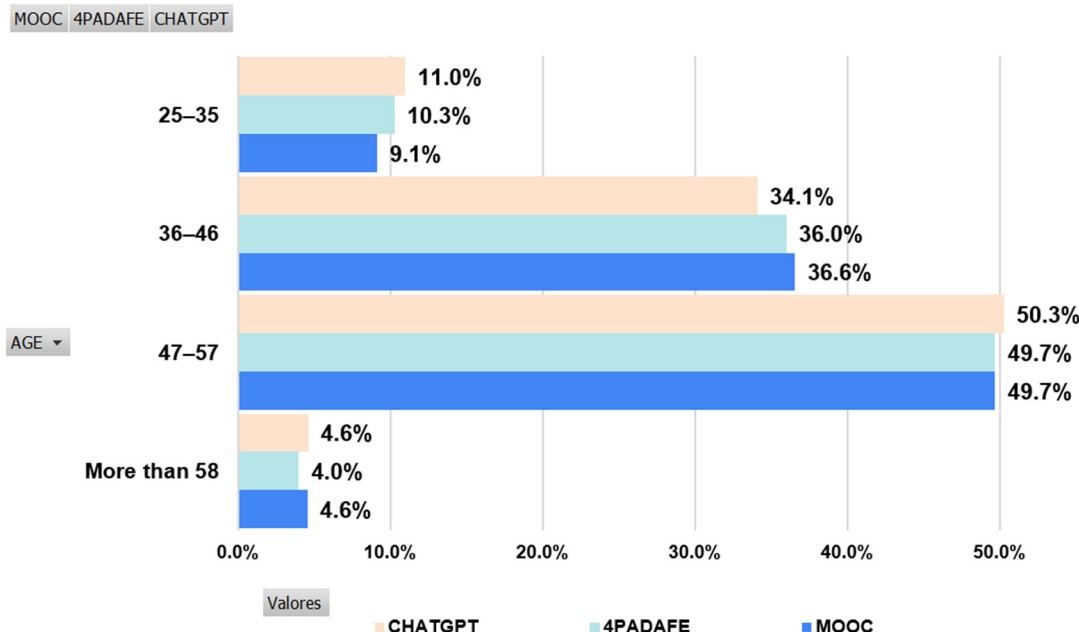

**Figure 7.** Summary of the survey responses of the 42 teachers.

Figure 8 illustrates that teachers perceive generative artificial intelligence tools as effectively enhancing students' learning motivation. The data reveal that teachers between the ages of 25 and 35 generally hold positive views, rating between 3 and 5 on the Likert scale; teachers aged 36 to 46 exhibit a range of opinions, with ratings between 2 and 5 on the Likert scale; and, although teachers aged 47 to 57 tend to be more negative, the majority still rate between 3 and 5 on the Likert scale, indicating a favorable perception overall.

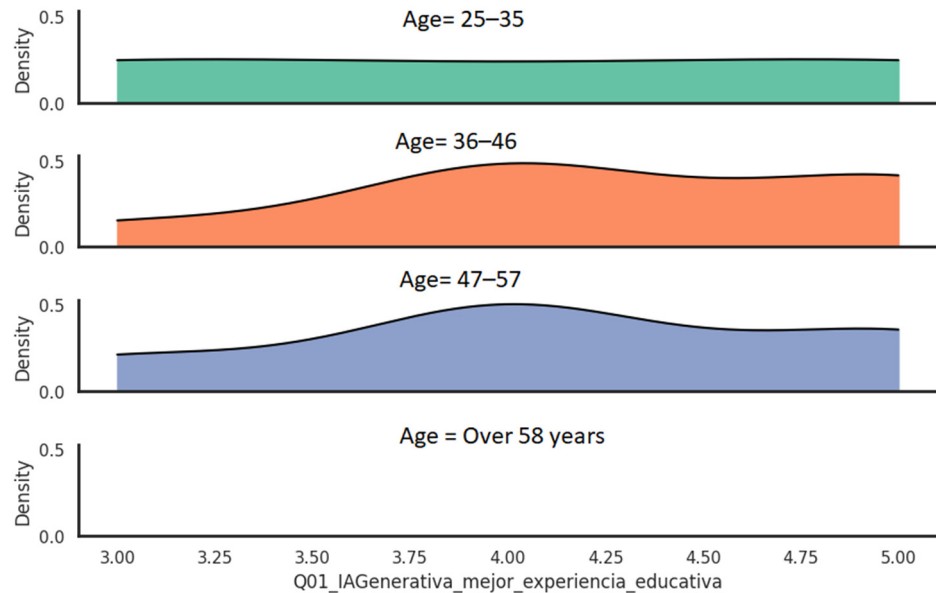

**Figure 8.** Generative artificial intelligence in conjunction with educational experience.

Figure 9 shows that teachers across all age ranges, from 25 to 58, hold positive opinions regarding the improvement of the educational experience through generative artificial intelligence tools. The scale ratings range between 3 and 5, demonstrating a widespread consensus among teachers regarding the positive impact of generative AI tools on the educational experience.

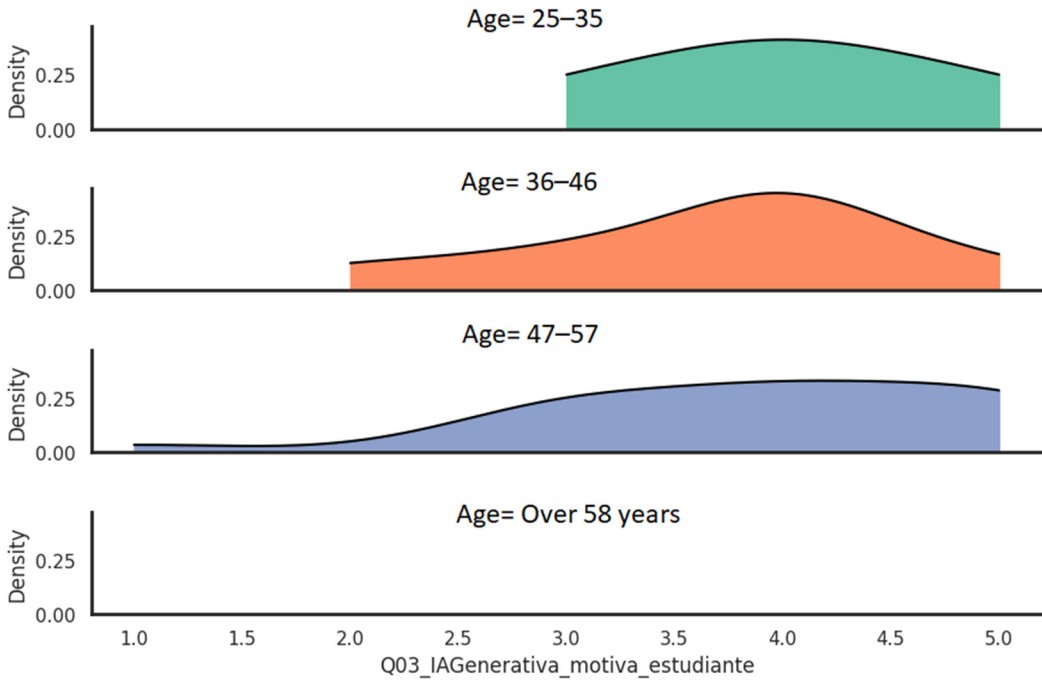

**Figure 9.** Generative artificial intelligence in conjunction with student motivation.

*Cluster Analysis and Modeling Groups with Random Forest*

Cluster analysis and group modeling with random forest have been widely used in data mining and machine learning. These techniques make it possible to identify patterns and group similar observations into distinct sets.

Cluster analysis, also known as clustering, is used to group objects or cases based on their similarity in characteristics or attributes. In the case of Figure 10, three clearly defined clusters can be observed. These clusters represent questions related to different aspects of the 4PADAFE methodology and the application of generative artificial intelligence in education.

Analysis of the quantitative data obtained from the survey identified several key themes related to participants' experiences and perspectives:

The participants highlighted the ability of these tools to personalize learning experiences, adapt to individual student needs, and enhance student motivation and engagement. The automation of content generation and the provision of real-time support were also recognized as significant benefits.

Some teachers expressed concerns about the reliability and accuracy of generative artificial intelligence tools, particularly in complex subject areas. Technical difficulties and the learning curve associated with utilizing these tools were also identified as challenges.

Participants emphasized the value of the 4PADAFE instructional design matrix in guiding the development of co-curricular activities. The structured approach provided clarity and coherence, ensuring that activities aligned with the learning objectives and promoted an effective educational process.

Integrating generative artificial intelligence tools and instructional design matrices significantly contributed to student engagement and active participation. Teachers reported feeling more connected to the course materials and appreciated the personalized support and tailored learning experiences.

These findings highlight the positive impact of generative artificial intelligence tools and instructional design matrices in constructing massive MOOC virtual classrooms. These tools and methodologies enhanced student engagement, personalized learning experiences, and improved educational outcomes.

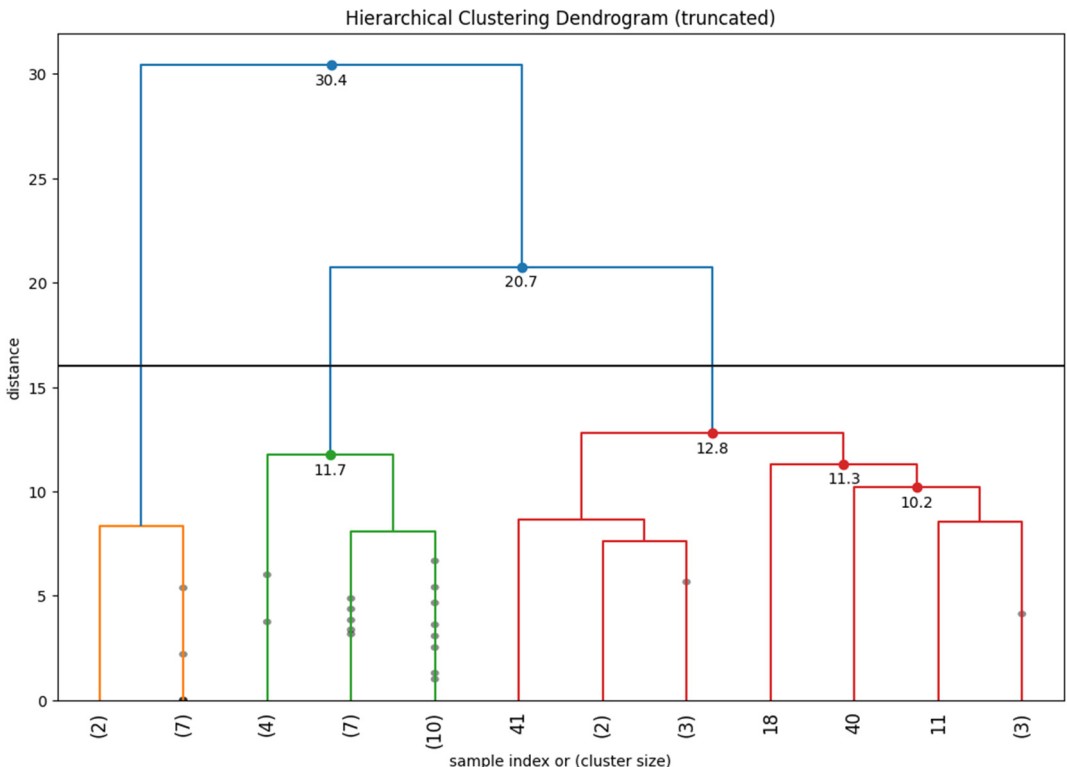

**Figure 10.** Summary of the most used generative AI tools.

Figure 11 summarizes the top ten questions, where red indicates strong disagreement, orange indicates disagreement, neutral green indicates agreement, light blue indicates agreement, and electric blue indicates strong agreement.

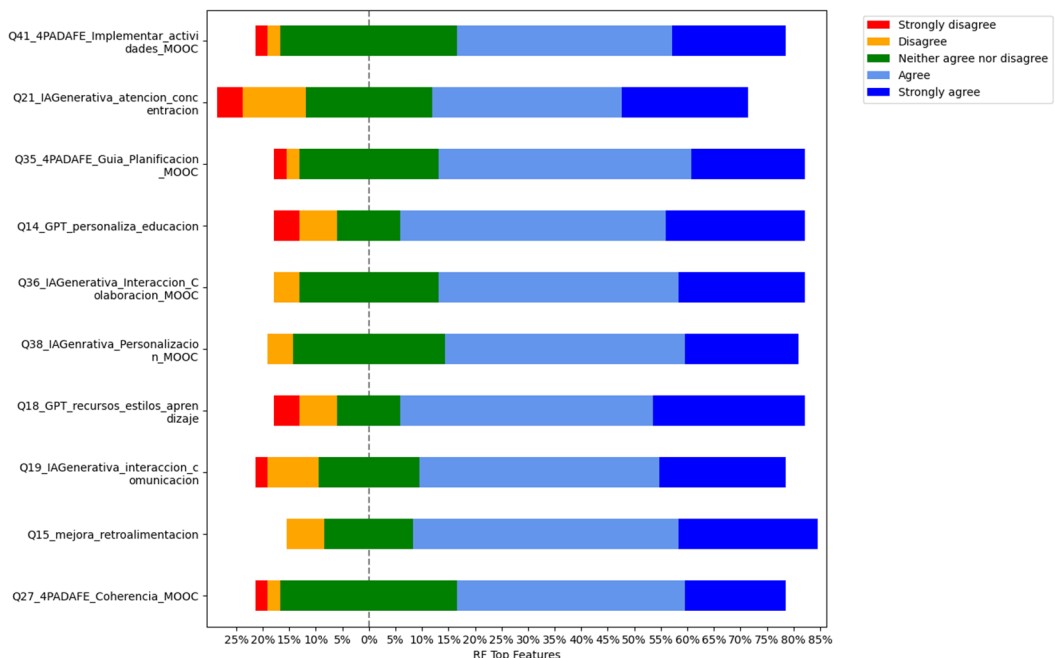

**Figure 11.** Summary of the top ten questions.

The study outcomes provide valuable insights for educators and institutions seeking to leverage technology and instructional design strategies to optimize online teaching and learning environments.

## 5. Discussion

The quantitative part of the research consists of applying the instructional design matrix of the 4PADAFE methodology in a virtual course called "Generative Artificial Intelligence Tools for Education: ChatGPT Techniques" The population included forty-two (42) teachers who participated in the course. A questionnaire of 45 questions was applied through Google Forms to be answered voluntarily.

The respondents were 50% women and 50% men; the teachers belonged to the different modalities of study, where the most significant number corresponded to the face-to-face modality with 59.5%, followed by the hybrid modality with 23.8%, the virtual modality with 14.3%, and the distance modality with 2.4%. Generative artificial intelligence tools and implementing an instructional design matrix are crucial in constructing massive virtual MOOC classrooms. These technological educational trends offer new opportunities to personalize learning, adapt to individual learner needs, and improve the quality of educational content. Generative artificial intelligence enables the automated creation of personalized educational materials, while the instructional design matrix provides structured guidance for designing and planning micro-curricular activities.

These technological trends become even more relevant in massive MOOC virtual classrooms aiming to reach many students. Generative artificial intelligence and the instructional design matrix make it possible to efficiently manage and organize educational content, adapting it to the needs of a diverse audience.

Teachers can use generative artificial intelligence tools to improve student engagement and motivation by providing relevant content tailored to their preferences and learning styles. In addition, implementing an instructional design matrix ensures consistency and quality in the design of activities by aligning them with educational objectives and providing a clear structure for their implementation.

Teachers value the potential support of these AI tools in the learning process, such as brainstorming and analysis. However, they also expressed concerns about privacy, ethics, and intellectual property.

Adopting generative artificial intelligence tools in education faces specific challenges, as a small percentage of teachers show total resistance to their use and do not recognize their relevance in the learning process. This resistance may be due to the lack of opportunities to experiment with these tools and the lack of alignment with current educational trends. To achieve successful implementation, it is critical to address the concerns and resistance of these teachers by providing them with the training and support necessary to understand the potential benefits of generative artificial intelligence tools in education.

The data analysis revealed that generative artificial intelligence tools and the instructional design matrix are powerful resources to improve the quality and effectiveness of MOOCs' massive virtual classrooms. These technological educational trends offer innovative opportunities to personalize learning [28], adapt to students' individual needs, and promote a more inclusive and accessible education. As we continue to explore and harness the potential of these tools, we can move toward a more inclusive and accessible education [29,30].

In addition, using generative artificial intelligence tools in education promotes inclusion and equity by improving access to information for students with different abilities and learning styles. The personalization of educational content and activities allows us to meet the specific needs of each student, facilitating their academic progress and comprehensive development.

Integrating virtual learning assistants based on generative artificial intelligence is also an effective strategy to enhance students' critical thinking and creativity. These assistants provide individualized support, stimulate problem-solving, and promote the exploration of new ideas, which contributes to the development of higher cognitive skills.

On the other hand, offering integrative curricula based on generative artificial intelligence tools provides students with a broader and multidimensional vision of knowledge,

fostering an understanding of interdisciplinary relationships and preparing them to face the challenges of the real world.

Using generative artificial intelligence tools in education has opened up new possibilities and opportunities to improve teaching and learning processes. However, it is essential to recognize and discuss the inherent limitations of these tools, including potential ethical issues, biases, and other limitations that may arise in their application in educational settings.

These tools collect and analyze large amounts of data, raising concerns about student information privacy and security. In addition, there is the potential for these tools to collect sensitive or personal data without proper consent, raising questions about compliance with data protection regulations and ethics in handling student information.

These tools are trained using existing data sets, which may contain inherent biases, such as gender, race, or social class biases. This can lead to biased or discriminatory results in generating content or the answers provided by the tools. Addressing these biases and ensuring fairness and impartiality in using generative artificial intelligence tools in education is critical.

In addition, technical limitations are also a factor to consider; although generative artificial intelligence tools have made significant progress in generating coherent and relevant content, they can still present difficulties in understanding complex contexts or interpreting abstract concepts. This can affect the quality and accuracy of the answers generated, which requires teacher supervision and correction to ensure the accuracy of the information provided to students.

Another aspect to consider is the over-reliance on generative artificial intelligence tools. While these tools can help provide additional support and facilitate the teaching process, there is a risk that students may become overly dependent on them and fail to develop critical skills, such as problem-solving or critical thinking. Balancing these tools with pedagogical approaches that foster students' holistic development is essential.

Applying generative artificial intelligence tools and implementing the 4PADAFE instructional design matrix is crucial for building massive virtual MOOC classrooms. These educational technology trends offer new opportunities to personalize learning, adapt to individual learner needs, and improve the quality of educational content. Generative artificial intelligence tools enable the automated creation of personalized educational materials, while the instructional design matrix provides structured guidance for designing and planning micro-curricular activities. These technology trends are especially relevant in massive MOOC virtual classrooms that seek to reach many students. Generative artificial intelligence and the instructional design matrix make it possible to efficiently manage and organize educational content, tailoring it to the needs of a diverse audience. Teachers can use generative artificial intelligence tools to improve student engagement and motivation by providing relevant content tailored to their preferences and learning styles. Implementing the instructional design matrix ensures consistency and quality in the design of the activities, aligning them with the educational objectives and providing a clear structure for performance.

One limitation of our study is the relatively small sample size of 42 teachers. This may limit the generalizability of the findings to larger populations. Future research could consider expanding the sample size to include a more diverse range of participants from different regions and academic disciplines to enhance the generalizability of the results.

Another limitation is the focus on specific generative artificial intelligence tools and instructional design matrices. Other tools and methodologies not included in our study may be available. Future research could explore a broader range of tools and methodologies to provide a more comprehensive understanding of their impact in virtual classrooms.

To gain deeper insights into teachers' experiences and perceptions, future research could incorporate qualitative data such as interviews or open-ended survey questions. This would provide a richer understanding of how teachers incorporate generative AI tools

in their teaching methods, specific tasks or interactions facilitated by these tools, and any observed benefits or challenges.

Conducting longitudinal studies over an extended period could provide a more comprehensive assessment of the long-term effects and sustainability of using generative AI tools and instructional design matrices in virtual classrooms.

Including comparison groups of teachers who do not use generative AI tools or instructional design matrices would allow for a more rigorous evaluation of the effectiveness and impact of these tools on student engagement, learning outcomes, and overall educational experience.

By addressing these limitations and exploring additional avenues for research, we can further refine our understanding of the potential of generative AI tools and instructional design matrices in education and identify strategies to optimize their implementation and effectiveness.

Finally, generative artificial intelligence tools facilitate the initial development of ideas and reflection on them, favoring the generation of creative and innovative solutions. Similarly, automated assessment and other assessment innovations enable more accurate and objective tracking of student progress, provide immediate feedback, and support data-driven decision-making by teachers.

## 6. Practical Implications

The results of this study have important practical implications for implementing generative artificial intelligence tools and instructional design matrices in virtual classrooms. The following are some specific recommendations for educators and institutions interested in adopting these technologies:

Training and professional development: Providing educators with adequate training in using generative artificial intelligence tools and applying instructional design matrices is critical. This study includes opportunities to acquire the technical skills and pedagogical knowledge necessary to use these tools effectively in the educational environment. Educators must understand how to take full advantage of the capabilities of generative AI tools and how to integrate them into their teaching practices effectively.

Personalization and customization: The results of this study highlight the ability of generative AI tools to personalize learning and adapt to the individual needs of learners. Educators should explore leveraging these capabilities to deliver more relevant and meaningful learning experiences.

This involves using generative AI tools to provide personalized content and feedback and to adapt teaching strategies to students' learning styles and preferences.

Curriculum design and planning: Implementing instructional design matrices can be an effective strategy to ensure consistency and quality in the design of activities in virtual classrooms. Educators should use these matrices as structured guides to develop activities aligned with the learning objectives, curricular content, and teaching strategies. This will help ensure that the activities are relevant and practical and promote an effective educational process.

Evaluation and feedback: Generative AI tools can be necessary for student assessment and feedback. Educators can use these tools to automate task assessment and provide immediate feedback to students. However, educators must monitor and verify the accuracy and reliability of AI-generated assessments. In addition, it is crucial to combine automated assessment with personalized, human feedback to provide students with a complete picture of their progress and areas for improvement.

Ethical and privacy considerations: Educators should be aware of ethical and privacy issues when adopting generative AI tools. Respecting student rights and privacy and addressing potential bias in AI-generated results is essential. Educators must understand and comply with policies and regulations related to student data use and ensure information security and confidentiality.

Continuous evaluation and improvement: Ongoing evaluation of the impact and effectiveness of generative AI tools and instructional design matrices is critical. Educators and institutions should collect student feedback and analyze data to understand how these technologies influence teaching and learning. This feedback and analysis can help identify areas for improvement, adjust teaching practices, and optimize the implementation of generative AI tools.

## 7. Conclusions

This work is relevant for educators, educational institutions, and practitioners interested in harnessing the potential of generative artificial intelligence tools and the 4PADAFE instructional design matrix for educational purposes. It is also valuable for researchers and academics seeking to explore the impact of these technologies in educational settings. The contribution of this work lies in providing a comprehensive and practical understanding of how generative artificial intelligence tools and instructional design matrices can optimize virtual classrooms and enhance the learning experience. The study presents original research on the impact of these technologies in constructing large-scale virtual classrooms and offers valuable insights into their implementation, benefits, and challenges. Users can enjoy several advantages by leveraging generative artificial intelligence tools and instructional design matrices. These include increased personalization of learning, tailored to individual learner needs, enhanced engagement and motivation, automated generation of instructional content, improved efficiency in content management and organization, consistent activity planning, and a clear structure for designing and developing instructional activities. Ultimately, this approach can significantly enhance the quality of the educational process and foster better learning outcomes in virtual environments.

This report presented general findings without specific details regarding the frequency of use, specific tasks, or interactions facilitated by ChatGPT. In future research, we will incorporate additional questions to address this limitation and better understand how teachers incorporate these tools into their teaching methods. Addressing privacy, ethics, and intellectual property concerns associated with using generative artificial intelligence tools in education is crucial. Further research and experimentation are necessary to maximize the benefits of generative artificial intelligence tools in education. Exploring ways to enhance inclusion and equity through these tools is essential, ensuring equitable access to information and tailoring educational content and activities to meet students' needs.

Moreover, we can develop additional strategies to integrate generative-artificial-intelligence-based virtual learning assistants, promoting students' critical thinking skills and creativity. Exploring more integrative approaches to curriculum design based on generative artificial intelligence tools can foster a multidimensional view of knowledge and prepare students for real-world challenges. It is crucial to continue researching and developing innovations in automated assessment and other assessment techniques that enable accurate and objective tracking of student progress, providing immediate feedback, and supporting data-driven decision-making by teachers. In future research, we will consider incorporating qualitative data to obtain a more complete and holistic understanding of the effects and implementation of generative artificial intelligence tools in education.

**Author Contributions:** Conceptualization, L.I.R.-R., P.A.-V., J.D.-M.-L. and M.G.-R.; methodology, L.I.R.-R. and P.A.-V.; validation, L.I.R.-R. and P.A.-V.; formal analysis, L.I.R.-R., P.A.-V. and M.G.-R.; investigation, L.I.R.-R., P.A.-V., J.D.-M.-L. and M.G.-R.; resources, P.A.-V. writing—original draft preparation, L.I.R.-R., P.A.-V., J.D.-M.-L. and M.G.-R.; writing—review and editing, L.I.R.-R., P.A.-V., J.D.-M.-L. and M.G.-R.; supervision, P.A.-V. and M.G.-R.; project administration, P.A.-V.; funding acquisition, P.A.-V. All authors have read and agreed to the published version of the manuscript.

**Funding:** This research was funded by the Universidad de Las Américas-Ecuador, as part of the internal research project INI.PAV.22.01, INI.PAV.22.02 and project INI.PAV.23.01.

**Institutional Review Board Statement:** Not applicable.

**Informed Consent Statement:** Not applicable.

**Data Availability Statement:** https://doi.org/10.17632/2kyksx8hty.1 (accessed on 23 June 2023).

**Conflicts of Interest:** The authors declare no conflict of interest.

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
