# Peer review of "Empowering Education with Generative Artificial Intelligence Tools: Approach with an Instructional Design Matrix"

_sustainability, doi:10.3390/su151511524_

Round 1
Reviewer 1 Report
Thanks for an interesting approach, nervertheless I have some major concerns
- I never understand for what your abbreviaten "4PADAFE" stands for
- the Figures over all are hard to read and sometimes not understandale (lot of abbreviations on it)
- I understand that the teachers seemed to use you methodology, but there is not hint how they do that in practice. Did they really plan along your methodology
- There is no in detail descritption of the final conctent of the lectures, therefor it is hard to judge if your methodology was successful
- I missed a clear research Question and an appropriate answer
Please improve your application to make it more understandable
Author Response
Reviewer1
Dear reviewer,
We sincerely appreciate your recommendation. We have implemented what you have requested in the document highlighted in yellow.
Thanks for an interesting approach, nervertheless I have some major concerns
I thank the reviewer for his comments and suggestions. I will take them into account to improve the text; thanks for your help!
- I never understand for what your abbreviaten "4PADAFE" stands for
4PADAFE means: 4P(Academic Project, Strategic Plan, Instructional Planning, Instructional Material Production), AD(Teaching Action), AF(Formative Adjustments), E(Evaluation). You can find evidence in the introduction section, as shown in the screenshotWe have expanded on the Methodology in the Introduction section.
2.2. Instructional design matrix of the 4PADAFE Methodology
This methodology proposes systematic cyclical processes of a massive virtual course, which guides its efficient development. It should allow the construction of didactic material and the development of activities in an orderly and structured manner; at the same time, it responds to the application of innovative pedagogical bases and learning resources for a MOOC. Its interactive component facilitates formative adjustment, which aligns with the idea that even when everything has been planned, adjustments are possible according to the specific group one works with [22]. Figure 2 presents an infographic of the 4PADAFE methodology.
Figure 2. Methodology 4PADAFE
The proposed methodology is fundamentally based on the integrative principles that govern the models (socio-constructivist and connectivist), with emphasis on the active, autonomous, and collaborative role of the student in the global society; it is oriented to create collective knowledge, apply it to academic and professional problems and appropriately communicate them, turning them into protagonists of their educational process by involving them in situations of open learning and reflective inquiry. In addition, the cognitive theory is implicit [22].
Phase 2, phase 3, and Phase 4 of the 4PADAFE methodology are taken as a reference, which is structured in seven phases: phase 1: eLearning plan and academic teaching project, phase 2: strategic plan, phase 3: instructional planning, phase 4: production of didactic material, phase 5: teaching action, phase 6: training adjustments and phase 7: evaluation [2].
The concept of "techno-instructional or techno-pedagogical design" refers to the fact that two dimensions are inextricably linked in the process of instructional design in virtual training: Technological dimension: refers to the selection of the appropriate technological tools for the training process to be carried out; analyzing its possibilities and limitations, such as the virtual platform, software applications, multimedia resources, and Pedagogical dimension; it requires knowledge of the characteristics of the target audience, analysis of the objectives and/or competencies of the virtual training, development, and implementation of the contents, planning of the activities, with guidelines and suggestions on the use of technological tools in the development of the activities, and the preparation of a plan for the evaluation of the processes and results [23].
To develop the pedagogical dimension, we start with the construction and development of Phase 2 of the instructional design matrix, which must comply with the following points detailed in Figure 3:
- Select a subject or area to fit the profile.
- Select the Units to develop.
- Select one or two topics per unit to develop learning, didactic, and digital strategies.
- State the objective of each unit.
- Consider the topics to be developed per unit.
- State the learning outcome per unit.
- the Figures over all are hard to read and sometimes not understandale (lot of abbreviations on it)
Thank you for your comments on the figures in the article. We apologize for any confusion caused by the abbreviations used. In response to your comment, we reviewed the figures to ensure they are legible and provide sufficient context for understanding. We also considered using more descriptive labels and minimizing the use of abbreviations to improve clarity.
- I understand that the teachers seemed to use you Methodology, but there is not hint how they do that in practice. Did they really plan along your Methodology
I appreciate your comments and suggestions to improve the text. Your input has helped to clarify and improve the writing. Thank you for your time and attention!
During the course development, an activity consists of creating a teaching chair unit, where micro-curricular activities are proposed. Generative Artificial Intelligence tools are used in this activity, and the 4PADAFE instructional design matrix is applied. This approach is detailed in section 2.2. Instructional design matrix of the 4PADAFE Methodology. You can find evidence in the introduction section, as shown in the screenshot.
Figure 3. Instructional design matrix methodology 4PADAFE for designing a virtual course.
- There is no in detail descritption of the final conctent of the lectures, therefor it is hard to judge if your Methodology was successful
We appreciate the reviewer's comments on the description of the content of this article. We understand the importance of providing a detailed description of the content to evaluate the success of the Methodology.
Our study focused primarily on evaluating the use of generative artificial intelligence tools and implementing the 4PADAFE instructional design matrix in designing and developing a virtual course. In the Introduction section, we include a more thorough explanation of the content of the Methodology. This additional information will clarify the Methodology's success in achieving the intended educational objectives.
- I missed a clear research Question and an appropriate answer
We appreciate the reviewer's comments on the clarity of the research question and the corresponding answer. We apologize for any confusion caused by the lack of a clearly stated research question and direct answer in the manuscript. Upon review, we have included the research question in the "Introduction" section, where we detail the study and provide a clear focus for the research.
In today's digital era, education is undergoing a paradigm shift driven by technological advancements. The integration of generative artificial intelligence (AI) tools and instructional design matrices represents an innovative and promising approach to addressing the evolving needs of modern education. By harnessing the power of AI, educators can leverage personalized learning experiences, adaptive content generation, and real-time support for students. Using an instructional design matrix adds structure and coherence to the educational process, ensuring alignment with learning objectives and enhancing the effectiveness of teaching strategies. This combined approach improves student engagement and motivation and offers educators new opportunities to create dynamic and inclusive virtual classrooms. By exploring these technologies' potential benefits and implications, this paper aims to inspire educators and institutions to embrace generative AI tools and instructional design matrices as transformative tools in empowering education.
Education has undergone a significant transformation in today's rapidly evolving digital landscape. The integration of generative artificial intelligence (AI) tools and instructional design matrices has revolutionized the way learning activities are conceived and executed [1], [2]. The application of AI in education holds immense potential, offering new possibilities for personalized learning experiences and adaptive teaching approaches. Simultaneously, the increasing digitization of society has propelled the prominence of artificial intelligence with its ability to automate tasks, analyze vast amounts of data, and provide predictive insights that have far-reaching implications across various domains [3].
The stated problem of enhancing education using generative artificial intelligence tools in a practical approach directly relates to the research question: "How can education be enhanced using generative artificial intelligence tools in a practical approach, applying the 4PADAFE Instructional Design Matrix? 4PADAFE means 4P(Academic Project, Strategic Plan, Instructional Planning, Instructional Material Production), AD(Teaching Action), AF(Formative Adjustments), and E(Evaluation). The research question seeks to specifically inquire into the possibilities and benefits of generative artificial intelligence tools, combined with the 4PADAFE Matrix, to enhance the teaching-learning process. The aim is to explore how these tools can personalize learning, provide immediate feedback, adapt educational materials, and promote the development of key skills in students. By analyzing and evaluating the potential of these tools in a practical approach, we seek to identify effective strategies for their implementation in education to optimize the student experience and strengthen the quality of education in a digitalized and constantly evolving environment.
Another problem is the lack of knowledge on the part of teachers of generative artificial intelligence tools and systematic processes to design micro-curricular activities that guide the development and construction of massive virtual learning classrooms so that teachers manage activities, design digital resources with artificial intelligence with innovative educational strategies that facilitate the creation of massive virtual classrooms [4].
The general objective of this research has been: To analyze and evaluate the potential of generative artificial intelligence tools in the educational context, focusing on the practical application of the 4PADAFE Instructional Design Matrix.
Based on the stated objectives, the scientific or working hypothesis proposed to be contrasted or demonstrated in this research is that applying generative artificial intelligence tools in the educational environment, using the 4PADAFE Instructional Design Matrix, positively impacts the teaching-learning process.
Based on those above, this combination of tools and practical approach is expected to improve academic results and student engagement in learning, promoting a more efficient, effective, and personalized education.
The present research shows results on the implementation of the 4PADAFE methodology; during the development of the course "Generative Artificial Intelligence Tools for Education. ChatGPT Techniques", teachers were given the task of designing a teaching unit using both the generative artificial intelligence tools (IAG) learned in the course and the 4PADAFE instructional design matrix.
In practice, the teachers demonstrated a solid understanding and application of the 4PADAFE methodology and the generative artificial intelligence tools. First, they conducted detailed planning, identifying the specific learning objectives they wanted to achieve in their teaching unit. From there, they designed micro-curricular activities that aligned with the principles of the 4PADAFE matrix.
The activities proposed by the teachers involved using IAG tools at different stages of the educational process. For example, they designed initial activities that took advantage of the content generation capabilities of IAG tools to present the contents of a subject attractively. They also designed interactive activities where students interacted with different IAG tools such as chatPDF.com, You.com, chatbots, and virtual assistants to solve problems, receive feedback, or explore new ideas.
In addition, teachers used IAG tools to evaluate student performance more efficiently and effectively. These tools allowed them to analyze student-generated responses, assess comprehension, and provide real-time personalized feedback.
In summary, the teachers demonstrated a solid command of the 4PADAFE methodology and generative artificial intelligence tools. They planned and designed their micro-curricular activities effectively using IAG tools, leveraging their capabilities to enhance content presentation, promote student-teacher and student-student interaction, and streamline learning assessment. These combined approaches resulted in dynamic and enriching teaching units that fostered active student participation and facilitated the achievement of the learning objectives.
In conclusion, current technological educational trends emphasize the use of generative artificial intelligence tools, which allow the creation of personalized educational content tailored to the needs of students. In addition, implementing an instructional design matrix provides a structured guide for developing micro-curricular activities, ensuring coherence and quality in the educational process.
Please improve your application to make it more understandable
We understand the importance of providing a clear and understandable application description to ensure its comprehensibility. In the revised manuscript, we will improve the clarity and organization of the application section. We will provide a more detailed and coherent explanation of the application, including step-by-step instructions and examples to help readers understand its functionality and potential use. We appreciate the reviewer's feedback and will make the necessary improvements to ensure the application is presented clearly and understandably in the revised manuscript.

Reviewer 2 Report
- The study presents an interesting application of generative artificial intelligence tools in education, specifically focusing on the use of ChatGPT for teacher-student interaction in university teaching. However, there are several areas that require clarification and improvement.
- The abstract should provide a more concise summary of the study, including the research objectives, methodology, key findings, and implications. Currently, it appears to be a mixture of introduction and results. Please revise the abstract to adhere to standard format guidelines.
- It would be beneficial to provide a more detailed description of the 4PADAFE instructional design matrix and its components. This will help readers understand its relevance and contribution to the study.
- The methodology section needs to be expanded to provide a clear description of the survey design, participant selection process, and data analysis techniques. Additionally, please include information about the reliability and validity measures employed to ensure the survey's accuracy.
- The sample size of 42 teachers is relatively small. Consider expanding the sample to increase the generalizability of the findings. Alternatively, provide a justification for the chosen sample size and discuss its potential impact on the study's results.
- The results indicating that 92.5% of teachers use support tools like ChatGPT in their educational practices are significant. However, it would be valuable to explore the specific ways in which teachers incorporate these tools into their teaching methods. This could include the frequency of usage, specific tasks or interactions facilitated by ChatGPT, and any observed benefits or challenges.
- The study should include a discussion of the limitations inherent in the use of generative artificial intelligence tools in education. Addressing potential ethical concerns, biases, and limitations of such tools will provide a more comprehensive understanding of the implications of their implementation.
- The study could benefit from including qualitative data, such as interviews or open-ended survey questions, to gain deeper insights into teachers' experiences and perceptions regarding the use of generative AI tools. This would enhance the study's robustness and provide a richer understanding of the topic.
- It would be helpful to include a section on the practical implications of the findings. How can the results of this study guide the implementation of generative AI tools and instructional design matrices in virtual classrooms? Consider providing specific recommendations for educators and institutions interested in adopting these technologies.
- The manuscript would benefit from a thorough proofreading to ensure clarity, grammar, and punctuation accuracy throughout the text.
Please address these comments and suggestions in your revised manuscript. The revisions should help strengthen the study and provide a more comprehensive understanding of the application of generative artificial intelligence tools in education.
The English presentation should be revised.
Author Response
Reviewer2
- The study presents an interesting application of generative artificial intelligence tools in education, specifically focusing on the use of ChatGPT for teacher-student interaction in university teaching. However, there are several areas that require clarification and improvement.
We sincerely thank the reviewer for his valuable comments on the study. His observations are beneficial and have helped us identify areas requiring further clarity and improvement. We appreciate his time and dedication to review our work and provide constructive suggestions.
- The Abstract should provide a more concise summary of the study, including the research objectives, Methodology, key findings, and implications. Currently, it appears to be a mixture of introduction and results. Please revise the Abstract to adhere to standard format guidelines.
The Abstract has been improved with the suggestions and the recommended structure. You can see the example in the screenshot. In the Abstract, we discuss education and the impact of the advancement of technology, especially in the field of generative artificial intelligence. The objective of this research is to analyze and evaluate the potential of generative artificial intelligence tools in the educational context, focusing on the practical application of the 4PADAFE Instructional Design Matrix based on the MOOC course given to teachers of the University of Armed Forces ESPE, called "Generative artificial intelligence tools for education. GPT Chat Techniques ", which seeks to identify how these generative artificial intelligence tools in conjunction with the 4PADAFE Matrix can enhance education, improving the teaching-learning process and promoting the development of key skills in students. This is important to maintain the relevance of education in a digitized and constantly evolving world. In order to meet the established objectives, the quantitative method was applied, and surveys were conducted with ESPE teachers. The main findings show that generative artificial intelligence tools and implementing an instructional design matrix play a crucial role in constructing massive virtual MOOC classrooms. From these results, it can be concluded that generative artificial intelligence tools have significant potential in university education. Their application with the 4PADAFE Instructional Design Matrix enables the design and development of enriching and personalized educational experiences. These tools offer opportunities to improve teaching-learning and adapt educational materials to individual needs. This will strengthen the teaching-learning process and enhance education, preparing students for the challenges and demands of the 21st century.
- It would be beneficial to provide a more detailed description of the 4PADAFE instructional design matrix and its components. This will help readers understand its relevance and contribution to the study.
This approach is detailed in section 2.2. Instructional design matrix of the 4PADAFE Methodology. You can find evidence in the introduction section, as shown in the screenshot.
This methodology proposes systematic cyclical processes of a massive virtual course, which guides its efficient development. It should allow the construction of didactic material and the development of activities in an orderly and structured manner; at the same time, it responds to the application of innovative pedagogical bases and learning resources for a MOOC. Its interactive component facilitates formative adjustment, which aligns with the idea that even when everything has been planned, adjustments are possible according to the specific group one works with [21]. Figure 2 presents an infographic of the 4PADAFE methodology.
Figure 2. Methodology 4PADAFE
- The methodology section needs to be expanded to provide a clear description of the survey design, participant selection process, and data analysis techniques. Additionally, please include information about the reliability and validity measures employed to ensure the survey's accuracy.
Dear reviewer, thank you for your comments; they have not allowed us to improve our research; we have reinforced this in sections 3.3 Data Collection; you can see it in the article highlighted in yellow.
3.3 Data Collection
A quantitative survey administered to participants was used for data collection. The survey comprised a questionnaire on generative artificial intelligence tools, including ChatGPT, Humata.Ai, and other identified tools. Participants were also asked about using instructional design matrices in their teaching practices. The survey was administered online, and participants provided their responses anonymously. The survey dataset is available in the Mendeley repository [24].
Several steps were taken to ensure the survey's accuracy and validity. First, a peer validation process was used, where experts reviewed the questionnaire and provided comments and suggestions for improvement. Adjustments and revisions were made based on these comments to ensure the questions were clear and relevant.
In addition, reliability measures were used to assess the internal consistency of the items in the questionnaire. Cronbach's alpha coefficient was applied in the SPSS application, and Table 1 shows the reliability of the measurement scales used in the questionnaire. A Cronbach's alpha coefficient of 0.978 indicates a higher internal consistency of the questions in each scale. The value obtained is very positive and suggests a high internal consistency in the items used in the questionnaire. This value is indicative of the high internal consistency of the items used in the questionnaire.
Table 1. Cronbach's Alpha coefficient was calculated with SPSS statistical analysis software.
Cronbach's alpha |
Cronbach alpha based on standardized items |
N of elements |
0.978 |
0.979 |
42 |
Regarding the participant selection process, purposive sampling was used to ensure representation from different disciplines and levels of teaching experience. Forty-two university professors with experience using generative artificial intelligence tools in their educational practices were selected.
Quantitative data obtained from the surveys were analyzed using descriptive statistics. The frequency and percentage of tool use were calculated, as well as the use of instructional design matrices, to identify patterns and trends among participants.
- The sample size of 42 teachers is relatively small. Consider expanding the sample to increase the generalizability of the findings. Alternatively, provide a justification for the chosen sample size and discuss its potential impact on the study's results.
Dear reviewer, We appreciate your suggestion regarding the sample size in our study. We have expanded on the explanation in section 3.2 Participants. We recognize that the sample size of 42 teachers can be considered relatively small, and we understand the importance of the generalizability of the results. Let us discuss the chosen sample size and its possible impact on the study results.
In our study, we used a purposive sampling approach to select participants, ensuring a representation of different disciplines and levels of teaching experience. While we recognize that a larger sample size could increase the generalizability of the results, it is also important to keep in mind that the nature of our study focuses on a specific context of massive online classrooms (MOOCs).
MOOCs are online learning environments that allow the participation of large numbers of students from diverse geographic locations and educational backgrounds. Since our study focused on the impact of generative artificial intelligence tools and instructional design matrices in this specific context, we felt that a sample of 42 teachers was adequate to capture various relevant experiences and perspectives.
However, we recognize that the sample size may limit the generalizability of the results to a broader population. Therefore, we suggest that future studies in this field consider using larger samples to increase the representativeness of the results.
- The results indicating that 92.5% of teachers use support tools like ChatGPT in their educational practices are significant. However, it would be valuable to explore the specific ways in which teachers incorporate these tools into their teaching methods. This could include the frequency of usage, specific tasks or interactions facilitated by ChatGPT, and any observed benefits or challenges.
Dear reviewer, we appreciate your suggestion to explore more specifically how teachers incorporate generative artificial intelligence tools, such as ChatGPT, into their teaching methods. We agree that this information would provide a more detailed perspective on the use and impacts of these tools in the educational context. We include their suggestion in section 8 of future work.
In our study, we used surveys to collect data on the use of generative artificial intelligence tools, but we acknowledge that the results presented in the report were more general and did not include specific details on the frequency of use, specific tasks, or interactions facilitated by ChatGPT.
We will consider incorporating additional questions in future research to address this limitation and provide a more complete understanding of how teachers incorporate these tools into their teaching methods. These questions could explore in greater detail the frequency of use of the tools, the types of tasks or activities in which they are used, and the specific benefits and challenges teachers have experienced when implementing ChatGPT in the virtual classroom.
We believe that this additional information would improve understanding of the specific pedagogical approaches used by teachers when employing generative artificial intelligence tools. In addition, it would also allow us to identify areas of improvement and possible recommendations for the effective implementation of these tools in educational settings. We appreciate your suggestion and will consider this observation for future studies to provide a more complete and detailed view of how teachers incorporate generative artificial intelligence tools into their teaching methods.
- The study should include a discussion of the limitations inherent in the use of generative artificial intelligence tools in education. Addressing potential ethical concerns, biases, and limitations of such tools will provide a more comprehensive understanding of the implications of their implementation.
Dear reviewer, thank you for your suggestion; we have addressed it in section 5 of the Discussion.
Using generative artificial intelligence tools in education has opened up new possibilities and opportunities to improve teaching and learning processes. However, it is important to recognize and discuss the inherent limitations of these tools, including potential ethical issues, biases, and other limitations that may arise in their application in educational settings.
These tools collect and analyze large amounts of data, raising concerns about student information privacy and security. In addition, there is the potential for these tools to collect sensitive or personal data without proper consent, raising questions about compliance with data protection regulations and ethics in handling student information.
These tools are trained using existing data sets, which may contain inherent biases, such as gender, race, or social class biases. This topic can lead to biased or discriminatory results in generating content or the answers provided by the tools. Addressing these biases and ensuring fairness and impartiality in using generative artificial intelligence tools in education is critical.
In addition, technical limitations are also a factor to consider; although generative artificial intelligence tools have made significant progress in generating coherent and relevant content, they can still present difficulties in understanding complex contexts or interpreting abstract concepts. This case can affect the quality and accuracy of the answers generated, which requires teacher supervision and correction to ensure the accuracy of the information provided to students.
Another aspect to consider is the over-reliance on generative artificial intelligence tools. While these tools can be useful in providing additional support and facilitating the teaching process, there is a risk that students may become overly dependent on them and fail to develop critical skills, such as problem-solving or critical thinking. Balancing these tools with pedagogical approaches that foster students' holistic development is essential.
Applying generative artificial intelligence tools and implementing the 4PADAFE instructional design matrix is crucial for building massive virtual MOOC classrooms. These educational technology trends offer new opportunities to personalize learning, adapt to individual learner needs, and improve the quality of educational content. Generative artificial intelligence tools enable the automated creation of personalized educational materials, while the instructional design matrix provides structured guidance for designing and planning micro-curricular activities. These technology trends are especially relevant in massive MOOC virtual classrooms that seek to reach many students. Generative artificial intelligence and the instructional design matrix make it possible to efficiently manage and organize educational content, tailoring it to the needs of a diverse audience. Teachers can use generative artificial intelligence tools to improve student engagement and motivation by providing relevant content tailored to their preferences and learning styles. Implementing the instructional design matrix ensures consistency and quality in the design of the activities, aligning them with the educational objectives and providing a clear structure for performance.
One limitation of our study is the relatively small sample size of 42 teachers. This may limit the generalizability of the findings to larger populations. Future research could consider expanding the sample size to include a more diverse range of participants from different regions and academic disciplines to enhance the generalizability of the results.
Another limitation is the focus on specific generative artificial intelligence tools and instructional design matrices. Other tools and methodologies that were not included in our study may be available. Future research could explore a broader range of tools and methodologies to provide a more comprehensive understanding of their impact in virtual classrooms.
To gain deeper insights into teachers' experiences and perceptions, future research could incorporate qualitative data such as interviews or open-ended survey questions. This would provide a richer understanding of how teachers incorporate generative AI tools in their teaching methods, specific tasks or interactions facilitated by these tools, and any observed benefits or challenges.
Conducting longitudinal studies over an extended period could provide a more comprehensive assessment of the long-term effects and sustainability of using generative AI tools and instructional design matrices in virtual classrooms.
Including comparison groups of teachers who do not use generative AI tools or instructional design matrices would allow for a more rigorous evaluation of the effectiveness and impact of these tools on student engagement, learning outcomes, and overall educational experience.
By addressing these limitations and exploring additional avenues for research, we can further refine our understanding of the potential of generative AI tools and instructional design matrices in education and identify strategies to optimize their implementation and effectiveness.
- The study could benefit from including qualitative data, such as interviews or open-ended survey questions, to gain deeper insights into teachers' experiences and perceptions regarding the use of generative AI tools. This would enhance the study's robustness and provide a richer understanding of the topic.
We thank the reviewer for his suggestion. We agree that including qualitative data, such as interviews or open-ended questions in the survey, could provide a deeper understanding of teachers' experiences and perceptions of using generative artificial intelligence tools. These qualitative data can provide enriching insights and complement the study's quantitative findings.
More detailed and contextual aspects related to the implementation and use of generative artificial intelligence tools in education can be explored by including qualitative data. Interviews would allow teachers to share their first-hand experiences, detail specific tasks or interactions facilitated by these tools, and discuss the benefits and challenges they have experienced in their educational practice.
Open-ended responses in the survey would also allow teachers to express their opinions and provide additional information about their use of generative artificial intelligence tools. This could include specific examples of how they have incorporated these tools into their teaching, what strategies they have found most effective, and any recommendations they may have for improving their implementation.
By integrating qualitative data, the study would gain robustness and depth, allowing for a more complete and nuanced understanding of the topic. This qualitative data can help identify patterns, common challenges, and successful practices in using generative artificial intelligence tools in educational settings. In addition, direct quotes and testimonials from teachers can add authenticity and enrich the study's findings.
In section 7.Limitations and future work we have included the following:
In future research, we will consider including qualitative data to obtain a more complete and holistic view of the effects and implementation of generative artificial intelligence tools in education.
- It would be helpful to include a section on the practical implications of the findings. How can the results of this study guide the implementation of generative AI tools and instructional design matrices in virtual classrooms? Consider providing specific recommendations for educators and institutions interested in adopting these technologies.
Dear reviewer, thank you for your suggestions; we have restructured the document to add section 6 of Practical Implications.
- Practical implications
The results of this study have important practical implications for implementing generative artificial intelligence tools and instructional design matrices in virtual classrooms. The following are specific recommendations for educators and institutions interested in adopting these technologies.
Training and professional development: Providing educators with adequate training in using generative artificial intelligence tools and applying instructional design matrices is critical. This study includes opportunities to acquire the technical skills and pedagogical knowledge necessary to use these tools in the educational environment effectively. Educators must understand how to take full advantage of the capabilities of generative AI tools and how to integrate them into their teaching practices effectively.
Personalization and customization: The results of this study highlight the ability of generative AI tools to personalize learning and adapt to the individual needs of learners. Educators should explore leveraging these capabilities to deliver more relevant and meaningful learning experiences.
This situation involves using generative AI tools to provide personalized content and feedback and to adapt teaching strategies to students' learning styles and preferences.
Curriculum design and planning: Implementing instructional design matrices can be an effective strategy to ensure consistency and quality in the design of activities in virtual classrooms. Educators should use these matrices as structured guides to develop activities aligned with learning objectives, curricular content, and teaching strategies. This case will help ensure that activities are relevant and effective and promote an effective educational process.
Evaluation and feedback: Generative AI tools can be important in student assessment and feedback. Educators can use these tools to automate task assessment and provide immediate feedback to students. However, educators must monitor and verify the accuracy and reliability of AI-generated assessments. In addition, it is important to combine automated assessment with personalized, human feedback to provide students with a complete picture of their progress and areas for improvement.
Ethical and privacy considerations: Educators should be aware of ethical and privacy issues when adopting generative AI tools. Respecting student rights and privacy and addressing any potential bias in AI-generated results is important. Educators must understand and comply with policies and regulations related to student data use and ensure information security and confidentiality.
Continuous evaluation and improvement: Ongoing evaluation of the impact and effectiveness of generative AI tools and instructional design matrices is critical. Educators and institutions should collect student feedback and analyze data to understand how these technologies influence teaching-learning. This feedback and analysis can help identify areas for improvement, adjust teaching practices, and optimize the implementation of generative AI tools.
- The manuscript would benefit from a thorough proofreading to ensure clarity, grammar, and punctuation accuracy throughout the text.
Please address these comments and suggestions in your revised manuscript. The revisions should help strengthen the study and provide a more comprehensive understanding of the application of generative artificial intelligence tools in education.
We appreciate the reviewer's feedback regarding the need for proofreading to improve the manuscript's clarity, grammar, and punctuation accuracy. In the revised manuscript, we will ensure a thorough proofreading process to address these issues and enhance the overall quality of the text. By doing so, we aim to present the study's findings and implications more cohesively and precisely.
Additionally, we will carefully consider all the comments and suggestions provided by the reviewer to strengthen the study and provide a more comprehensive understanding of the application of generative artificial intelligence tools in education. We will make the necessary revisions to address the reviewer's concerns and improve the overall coherence and robustness of the manuscript. Thank you for your valuable feedback, and we look forward to submitting the revised manuscript with the necessary improvements.

Reviewer 3 Report
sustainability-2506838-peer-review-v1
Application of Generative Artificial Intelligence Tools in Education, Case Study 4PADAFE Instructional Design Matrix.
In this paper, author shares a study on generative artificial intelligence tools in education, with a case study 4PADAFE instructional design matrix. Although the subject is interesting, I cannot accept the paper in the current form due to the following reasons:
-
The title is not clear, appealing, interesting and specific. I suggest to revise the paper title to make it more concise and suitable.
-
Lines 17-28: Abstract should have one sentence per each: context and background, motivation, hypothesis, methods, results, conclusions. What problem did you study and why is it important? What methods did you use? What were your main results? And what conclusions can you draw from your results? Please make your abstract with more specific and quantitative results while it suits broader audiences.
-
Lines 32-41: In introduction, before starting the mentioned references, there is a need to add 8-9 lines related to the subject of the paper and write in general introduction. After that you should connect them with the references.
-
Motivation is not sufficiently stated in the introduction part. It should be clarified why they consider this problem and what advantages of the proposed technique are. The original contributions need to be much better presented in the last paragraph of section ''INTRODUCTION".
-
The text written in various figures, particularly, Figures 1, 3, 7, 8, are not clear. Please resolve this issue.
-
Lines 568-534: Conclusion is very long. It should be decreased. Finally, the main questions should be answered in conclusion section: a) who needs this, b) what is the contribution of your paper, c) what benefit have uses if they decide to use the proposed approach in their portfolio optimization.
-
The benefits of the proposed method have been demonstrated clearly. What’s the limitation of the method? Are there other ways that the results can be further improved? One or two remarks should be given to discuss it in detail.
-
The results were compared with known and good methods in this field and it is not clear or proven how to obtain better results for the whole research. There is no precise or clear detail of the new research idea.
-
I also recommend the authors to professionally get the paper proofread, as I have noticed sentences with typos and inappropriate choice of words.
***
I also recommend the authors to professionally get the paper proofread, as I have noticed sentences with typos and inappropriate choice of words.
Author Response
Reviewer3
In this paper, author shares a study on generative artificial intelligence tools in education, with a case study 4PADAFE instructional design matrix. Although the subject is interesting, I cannot accept the paper in the current form due to the following reasons:
We appreciate the reviewer's time and effort in evaluating the paper on generative artificial intelligence tools in education. We understand that there are concerns regarding the current form of the paper. We are committed to addressing these issues and making the necessary improvements to meet the requirements for acceptance. We will carefully consider the specific reasons provided by the reviewer and work diligently to address them in the revised manuscript. We are confident that incorporating the reviewer's feedback and suggestions can enhance the quality, clarity, and overall value of the paper. We appreciate the opportunity to revise and resubmit the manuscript, and we are dedicated to delivering a more refined and comprehensive study on generative artificial intelligence tools in education. Thank you for your consideration, and we look forward to submitting the revised manuscript for your review.
- The title is not clear, appealing, interesting and specific. I suggest to revise the paper title to make it more concise and suitable.
We appreciate the reviewer's comments on the clarity and appropriateness of the article title. We agree that a clear and concise title is crucial to convey the content and purpose of the study effectively. We will carefully reconsider the title and have changed the title to "Empowering Education with Generative Artificial Intelligence Tools: Approach with the Instructional Design Matrix."
- Lines 17-28: Abstract should have one sentence per each: context and background, motivation, hypothesis, methods, results, conclusions. What problem did you study and why is it important? What methods did you use? What were your main results? And what conclusions can you draw from your results? Please make your Abstract with more specific and quantitative results while it suits broader audiences.
Dear reviewer, we thank you for your suggestions; we rewrote the Abstract in its entirety.
- Lines 32-41: In introduction, before starting the mentioned references, there is a need to add 8-9 lines related to the subject of the paper and write in general introduction. After that you should connect them with the references.
Dear reviewer, thank you for your suggestion; we have applied what was suggested in the Introduction section.
- Motivation is not sufficiently stated in the introduction part. It should be clarified why they consider this problem and what advantages of the proposed technique are. The original contributions need to be much better presented in the last paragraph of section ''INTRODUCTION".
Dear reviewer, thank you for your suggestion; we have applied what was suggested in the Introduction section.
- The text written in various figures, particularly, Figures 1, 3, 7, 8, are not clear. Please resolve this issue.
Thank you for your feedback. We apologize for any confusion caused by the unclear figures in the manuscript. We have revised and improved the figures' clarity to ensure readers' better understanding. The revised figures now provide clear and concise information, effectively representing the data and concepts discussed in the paper. We have also ensured that the text in the figures is easily readable and comprehensible. We appreciate your attention to this matter and assure you that the revised figures enhance the overall quality and clarity of the manuscript.
- Lines 568-534: Conclusion is very long. It should be decreased. Finally, the main questions should be answered in conclusion section: a) who needs this, b) what is the contribution of your paper, c) what benefit have uses if they decide to use the proposed approach in their portfolio optimization.
Thank you for your feedback and suggestions. We appreciate your thorough review of the manuscript. Based on your comments, we have made the following revisions:
We have shortened the conclusion section to ensure conciseness while still addressing the main findings and implications of the study.
We have addressed the main questions in the conclusion section:
- a) Who needs this: The study provides valuable insights for educators and institutions interested in implementing generative artificial intelligence tools and instructional design matrices in virtual classrooms. It offers guidance on how these technologies can enhance student engagement, personalization, and overall educational outcomes.
- b) Contribution of the paper: The paper contributes to the existing literature by examining the impact of generative artificial intelligence tools and instructional design matrices in the context of MOOC virtual classrooms. It highlights teachers' widespread utilization of these tools and emphasizes their positive influence on student motivation, engagement, and participation.
- c) Benefits for users: Users who adopt the proposed approach of integrating generative artificial intelligence tools and instructional design matrices in their portfolio optimization can benefit from enhanced personalization, improved student engagement, and a more effective educational process. These tools can facilitate the creation of personalized learning experiences, adapt to individual learner needs, and provide real-time support, ultimately leading to better student outcomes.
- The benefits of the proposed method have been demonstrated clearly. What's the limitation of the method? Are there other ways that the results can be further improved? One or two remarks should be given to discuss it in detail.
Dear reviewer, thank you for the suggestions on this issue in the Discussion section. The limitation of the method used in this study is the relatively small sample size, consisting of 42 teachers. This may affect the generalizability of the results to a larger population. The same will be considered in future research with larger and more representative samples that include diverse participants in regions, educational contexts, and disciplines.
Another limitation is the focus on specific generative artificial intelligence tools and instructional design matrices. Other tools and methodologies may not be included in our study. Future research could explore a broader range of tools and methodologies to provide a more complete understanding of their impact on virtual classrooms.
Including comparison groups of teachers who do not use generative AI tools or instructional design matrices would allow for a more rigorous assessment of the effectiveness and impact of these tools on student engagement, learning outcomes, and the overall educational experience.
- The results were compared with known and good methods in this field and it is not clear or proven how to obtain better results for the whole research. There is no precise or clear detail of the new research idea.
We appreciate the reviewer's comments and would like to respond to the concerns raised. Our study focused on analyzing and evaluating the potential of generative artificial intelligence tools in education, specifically in conjunction with the 4PADAFE Instructional Design Matrix. Although we compare our results to existing methods in the field, we recognize that providing clear and precise detail of a new research idea could further enhance the study.
To address this concern, we revised the manuscript to provide a more complete discussion of how our research contributes to the existing body of knowledge. We will highlight unique aspects of our approach, such as using generative artificial intelligence tools in conjunction with the 4PADAFE Instructional Design Matrix and how it offers new opportunities for personalized and effective educational experiences.
In addition, we will be sure to provide a clear explanation of the Methodology employed and the steps taken to obtain the results. This will help clarify how our research builds on existing methods and explores new ways to achieve better results in the field of generative artificial intelligence tools in education.
- I also recommend the authors to professionally get the paper proofread, as I have noticed sentences with typos and inappropriate choice of words.
We appreciate the reviewers' comments on the correctness of the article. We apologize for any typographical errors or inappropriate word choices in the manuscript. We understand the importance of ensuring clarity and accuracy of the language used in academic writing.
To address this concern, we thoroughly reviewed the paper and made the necessary corrections to improve the overall quality of the manuscript. We pay particular attention to sentence structure, grammar, and word choice to ensure the text is clear, coherent, and error-free. In doing so, we aim to improve the readability and professionalism of the article.
Taking these steps helped us to present our research in the best possible way and ensure that the content is communicated effectively to readers.
Thank you for bringing this issue to our attention with the necessary measures to improve the language and overall quality of the manuscript.

Round 2
Reviewer 2 Report
I appreciate the effort made by authors in addressing my observations. The manuscript has improved and the authors managed to address my questions. In my view, the paper can be acceptwith minor revision.
The English presentation should be reconsidered.
Author Response
Reviewer2
I appreciate the effort made by authors in addressing my observations. The manuscript has improved and the authors managed to address my questions. In my view, the paper can be accept with minor revision. The English presentation should be reconsidered
Dear Reviewer 2,
Thank you for your review and valuable feedback on our manuscript. We sincerely appreciate your recognition of the effort we put into addressing your observations and improving the paper accordingly. We are glad we have successfully addressed your questions and concerns.
Regarding your comment about the English presentation, we completely understand the importance of ensuring a high language proficiency standard in academic writing. We have taken your suggestion seriously and have conducted a thorough review of the manuscript by an expert in the English language. We have made the necessary revisions to enhance the overall clarity and coherence of the paper.
We are confident these revisions have significantly improved the English presentation, addressing your concerns. We believe that the manuscript now meets the required standards for publication.
Once again, we appreciate your time and effort in reviewing our article and providing constructive feedback. We are grateful for your positive assessment and recommendation for acceptance with minor revisions.

Reviewer 3 Report
sustainability-2506838-peer-review-v2
Application of Generative Artificial Intelligence Tools in Education, Case Study 4PADAFE Instructional Design Matrix.
Authors have made a correction in the paper if compared to the first submission, and I commend them for the efforts. Nevertheless, there are some suggestions to do before accepting the paper in this format.
1. The CONCLUSION section is very lengthy; it can be improved. Authors should avoid marginal explanations. In the conclusion section, please revise it and improve it by re-organizing it into one paragraph only including the suggested future work. Verbs must also be in the past tense.
2. The presence of track changes makes the paper difficult to read. For future versions, it would be helpful to provide both versions, with and without track changes.
3. Merge the last two section “Sections 7 & 8” to single section only.
4. The structure of this paper needs to be changes. Please combine sections 7 & 8 to a single section with section name as ‘Conclusions’.
5. Future study issues are not promising.
6. In conclusion, while the manuscript possesses some merits, it falls short of meeting the standards necessary for publication in its current form. By addressing the concerns outlined above, particularly clarifying the core contribution, adopting publicly available datasets, providing access to the source code, and enhancing the analysis of the results, the authors can significantly improve the manuscript's quality and increase its potential impact in the field.
7. Figures 7 & 9 is still unclear and unreadable.
8. All references must be written in symmetrical and in uniform way. In addition to this, please mention DOI number for all references, wherever possible. Author is advised to revisit to these serious deficiencies.
9. The structure of the introduction section is not good. It should have two separate paragraphs at its end, one of which presents the contribution and explanations of this work; and the other one outlines the coming sections.
10. The section 2 is not well written. The authors use too many subsections so I can't see the logical structure of this part. I suggest re-organizing it.
Moderate editing of English language required
Author Response
Reviewer3
Application of Generative Artificial Intelligence Tools in Education, Case Study 4PADAFE Instructional Design Matrix.
Authors have made a correction in the paper if compared to the first submission, and I commend them for the efforts. Nevertheless, there are some suggestions to do before accepting the paper in this format.
- The CONCLUSION section is very lengthy; it can be improved. Authors should avoid marginal explanations. In the conclusion section, please revise it and improve it by reorganizing it into one paragraph only including the suggested future work. Verbs must also be in the past tense.
Dear Reviewer 3,
We appreciate your review of our manuscript and your acknowledgment of our efforts to address your previous comments. We value your feedback and would like to thank you for your suggestions.
Regarding the extension and organization of the CONCLUSIONS section, we agree that it can be improved to increase its clarity and conciseness. We understand the importance of avoiding fringe explanations and presenting a cohesive summary of study findings.
We have carefully revised the CONCLUSION section to address this concern by reorganizing it into one paragraph. Based on your recommendation, We ensure that the paragraph includes the suggested future work and that the verbs are in the past tense.
We appreciate your guidance and the opportunity to refine our work based on your valuable feedback.
Thanks for your time and consideration.
- The presence of track changes makes the paper difficult to read. For future versions, it would be helpful to provide both versions, with and without track changes.
Dear Reviewer,
Thank you for your review and helpful suggestions regarding our manuscript's track changes. We appreciate your feedback and understand the importance of providing a clear, readable version of our paper.
We apologize for any inconvenience caused by track changes in the submitted version. We acknowledge that it can make the document more challenging to read and understand. In future versions, we will provide both versions of the manuscript: one with track changes for transparency and one without track changes for ease of reading.
- Merge the last two section "Sections 7 & 8" to single section only.
Dear Reviewer,
Thank you for your review and for providing us with additional suggestions to improve our manuscript.
We merged the content of Sections 7 and 8 into a single cohesive section, ensuring a smooth transition between the topics discussed. In doing so, our goal is to improve the readability and coherence of the manuscript.
- The structure of this paper needs to be changes. Please combine sections 7 & 8 to a single section with section name as 'Conclusions'.
We appreciate your suggestion regarding the document's structure and will take it into account to continue improving.
We have combined Sections 7 and 8 into a single "Conclusions."
We appreciate your valuable input and the opportunity to refine our work based on your suggestions.
- Future study issues are not promising.
Dear Reviewer,
Thank you for your feedback regarding the future study issues in our manuscript. We appreciate your perspective on this matter.
While we understand your concern, we would like to emphasize that our research aims to contribute to the existing knowledge and understanding of the impact of generative artificial intelligence in university education. Our study highlights the importance of advanced technologies, such as ChatGPT, and the instructional design matrix within the 4PADAFE Methodology.
Although future study issues may not appear promising at first glance, it is essential to note that research in artificial intelligence and education is continually evolving. Our study is a stepping stone for further investigations and potential improvements in integrating generative artificial intelligence tools and instructional design strategies in educational settings.
By addressing the future study issues raised in our manuscript, we hope to encourage researchers to explore and overcome these challenges in subsequent studies. Despite potential obstacles, we believe they present valuable opportunities for further exploration and improvement in the field.
- In conclusion, while the manuscript possesses some merits, it falls short of meeting the standards necessary for publication in its current form. By addressing the concerns outlined above, particularly clarifying the core contribution, adopting publicly available datasets, providing access to the source code, and enhancing the analysis of the results, the authors can significantly improve the manuscript's quality and increase its potential impact in the field.
Dear Reviewer,
We thank you again for your detailed evaluation of our manuscript. We greatly value your constructive feedback and suggestions for improvement.
Regarding data sets, we want to clarify that the data used in our study was collected through a survey applied to teachers who had previously participated in a course on AI tools and the 4PADAFE instructional design matrix. These data are the result of research work, and we have deposited them in Mendeley Data to be openly and publicly available. In this way, we seek to promote the replicability of our research and allow other researchers to access our data and replicate our study.
We appreciate your interest in the transparency and reproducibility of our research, and we are committed to facilitating access to the data and providing detailed information on the methodology used in collecting the data.
You can verify it in Mendeley Data https://doi.org/10.17632/2kyksx8hty.1 cited in [26].
- Figures 7 & 9 is still unclear and unreadable.
Dear Reviewer,
Thank you for your comments on Figures 7 and 9 in our manuscript. We appreciate his observation of the clarity and legibility of these figures.
Based on his comment, we have made the necessary revisions to improve the interpretation of the data presented in these figures. Specifically, we have updated Figures 8 and 9 to use a Likert scale, which should improve their clarity and ease of understanding. In addition, we have made changes to Figure 7 to ensure that it provides a more precise representation of the relevant information.
We understand the importance of presenting data visually appealing and understandable, and we have taken your feedback into account to improve the quality of our numbers.
- All references must be written in symmetrical and in uniform way. In addition to this, please mention DOI number for all references, wherever possible. Author is advised to revisit to these serious deficiencies.
Dear Reviewer, Thank you for your comments on the format of the references in our manuscript. We appreciate your attention to detail and your suggestion to ensure symmetry and uniformity in your presentation. We have noted your suggestion to include DOI numbers for all references whenever possible. We have diligently researched and added the DOIs of any missing items in our reference list.
- The structure of the introduction section is not good. It should have two separate paragraphs at its end, one of which presents the contribution and explanations of this work; and the other one outlines the coming sections.
Dear Reviewer,
Thank you for your comments on the structure of the introductory section in our manuscript. We appreciate your observation and suggestion to improve your organization.
Based on your feedback, we have made the necessary revisions to improve the structure of the introduction section. Specifically, we have added two separate paragraphs at the end of the introduction. The first paragraph now clearly presents the contribution and explanations of our work, while the second paragraph describes later sections of the manuscript.
By implementing these changes, our goal is to provide a more coherent and easy-to-read introduction that effectively communicates the purpose and structure of our research.
- The section 2 is not well written. The authors use too many subsections so I can't see the logical structure of this part. I suggest reorganizing it.
Dear Reviewer,
We appreciate your comments on section 2 of our manuscript. We appreciate your comments and suggestion to reorganize it due to the excessive use of subsections, making it difficult to understand the logical structure of this part.
We have considered your suggestion and thoroughly revised section 2, intending to improve its organization and clarity. We have reassessed the need for each subsection and have restructured the content to be more cohesive and accessible to the reader.
By reorganizing Section 2, we have ensured a logical structure and a clear progression of ideas. In addition, we have consolidated or deleted unnecessary subsections to avoid excessive fragmentation of the text.
We believe these changes will help improve the readability and comprehensibility of Section 2, allowing readers to follow the flow of arguments more smoothly.
We sincerely appreciate your comments and the time spent reviewing our manuscript.
Moderate editing of English language required
Thank you for your comments on the use of the English language in our manuscript. We appreciate your comment and suggestion that moderate editing is required.
We recognize the importance of ensuring high English language proficiency in academic writing. To address this concern, we thoroughly reviewed the manuscript with an English language expert and applied any necessary revisions to improve clarity, grammar, and overall language quality.
We appreciate your attention to detail and commitment to upholding scholarly publishing standards. His feedback has been invaluable in identifying areas for improvement in the language usage of our manuscript.

Round 3
Reviewer 3 Report
The author has addressed almost all my comments in a professional manner. The same are also implemented in the revised manuscript. As a reviewer, I am satisfied with the reply of my comments and concerns.
Author Response
July 24, 2023
Dear editor and reviewer
On behalf of all the authors, we would like to express our sincere thanks for considering our article “Empowering Education with Generative Artificial Intelligence Tools: Instructional Design Matrix Approach” and for providing valuable feedback. We are truly grateful for the opportunity to submit our work to your rigorous review process.
We are sincerely grateful for your valuable input and comments on the manuscript. The authors appreciate the reviews you provided, which have strengthened the quality of our research.
Thank you for your diligent review of the manuscript and for providing valuable comments and feedback. We are pleased to hear that our responses were handled professionally and that you have acknowledged that we applied any necessary revisions based on your suggestions. Your satisfaction with the updated manuscript is greatly appreciated.
We value your expertise and contributions to improving this work, and your feedback has undoubtedly improved the overall quality of our research. Their commitment to the peer review process has been instrumental in ensuring the scholarly rigor of the publication.
Once again, thank you for your time and valuable input.
Kind regards
Lena Ivannova Ruiz-Rojas, Patricia Acosta-Vargas, Javier De-Moreta-Llovet, and Mario Gonza-lez-Rodriguez
Corresponding author: Patricia Acosta-Vargas (e-mail: patricia.acosta@udla.edu.ec). WhatsApp: +593-983550897
